# SuRe: Summarizing Retrievals using Answer Candidates for Open-domain QA of LLMs

**Jaehyung Kim**[1,*]  **Jaehyun Nam**[2]  **Sangwoo Mo**[3]  **Jongjin Park**[2]
**Sang-Woo Lee**[2,5]  **Minjoon Seo**[2]  **Jung-Woo Ha**[4,5]  **Jinwoo Shin**[2]
[1]Carnegie Mellon University [2]KAIST AI [3]University of Michigan [4]Naver AI Lab [5]Naver Cloud
jaehyun4@andrew.cmu.edu

## ABSTRACT

Large language models (LLMs) have made significant advancements in various natural language processing tasks, including question answering (QA) tasks. While incorporating new information with the retrieval of relevant passages is a promising way to improve QA with LLMs, the existing methods often require additional fine-tuning which becomes infeasible with recent LLMs. Augmenting retrieved passages via prompting has the potential to address this limitation, but this direction has been limitedly explored. To this end, we design a simple yet effective framework to enhance open-domain QA (ODQA) with LLMs, based on the summarized retrieval (SuRe). SuRe helps LLMs predict more accurate answers for a given question, which are well-supported by the summarized retrieval that could be viewed as an explicit rationale extracted from the retrieved passages. Specifically, SuRe first constructs summaries of the retrieved passages for each of the multiple answer candidates. Then, SuRe confirms the most plausible answer from the candidate set by evaluating the validity and ranking of the generated summaries. Experimental results on diverse ODQA benchmarks demonstrate the superiority of SuRe, with improvements of up to 4.6% in exact match (EM) and 4.0% in F1 score over standard prompting approaches. SuRe also can be integrated with a broad range of retrieval methods and LLMs. Finally, the generated summaries from SuRe show additional advantages to measure the importance of retrieved passages and serve as more preferred rationales by models and humans.[1]

## 1 INTRODUCTION

Large language models (LLMs) (Brown et al., 2020; Touvron et al., 2023b) have significantly accelerated progress in natural language processing (NLP) and have become a core technology in various real-world applications used by millions of users, such as coding assistants (Chen et al., 2021), search engines (Xuan-Quy et al., 2023), and chatbots (Kim et al., 2021; OpenAI, 2022). However, LLMs often suffer from limitations, such as non-factual but seemingly plausible generation, referred to as hallucinations (Welleck et al., 2020), and difficulty in integrating up-to-date knowledge, as their learned knowledge is limited by the training corpus encoded in their parameters (Guu et al., 2020). This problem is particularly critical for question answering (QA) (Kwiatkowski et al., 2019), one of the most frequently encountered applications for LLMs.

Incorporating new information through the retrieval of relevant knowledge for a given query (*e.g.*, a question from users) is widely explored to improve the accuracy of QA systems, called open-domain QA (ODQA) (Karpukhin et al., 2020), and shows promise in addressing the aforementioned limitations of LLMs (Mialon et al., 2023). Constructing these *retrieval-augmented* LLMs typically involves additional fine-tuning (Borgeaud et al., 2022; Izacard et al., 2023), but it becomes infeasible due to the increase in scale and the recent nature of black-box API (OpenAI, 2023). Consequently, retrieval augmentation via *prompting*, *i.e.*, giving specific instruction as the input to obtain the desired outputs by LLM, becomes an attractive direction from its simplicity and efficiency (Shi et al., 2023). However, naïve prompting could be limited in fully exploiting the retrieved contexts, since

---

[*] This work is done when Jaehyung Kim was in KAIST.

[1]The code is available at https://github.com/bbuing9/ICLR24_SuRe

LLMs are simply instructed to use the retrieved information, instead of being explicitly trained to use it; for example, Liu et al. (2023b) recently observed that LLMs struggle to handle long input contexts when they are naïvely appended. Despite its importance, how to improve retrieval-augmented LLMs via prompting has been under-explored. Therefore, to improve ODQA via LLMs, we aim to develop a simple yet effective framework based on prompting, that could be easily applicable to various LLMs and retrieval methods.

**Contribution.** We propose a framework based on **Su**mmarized **Re**trieval (**SuRe**), to improve ODQA performance of retrieval-augmented LLMs. At a high level, SuRe helps LLMs predict more grounded answers, which are well-supported by the summarization of retrieved passages that could be viewed as an explicit rationale extracted from the retrieved passages. To be specific, SuRe first constructs the multiple summarizations of retrieved passages conditioned on each of a few possible answer candidates. It enables LLMs to focus on the specific contexts relevant to the given candidate, and hence provides more discriminative viewpoints for the given question. Then, using the generated summarizations, SuRe confirms the most plausible answer among candidates by measuring the corresponding summaries' validity to support the given candidate and ranking of relative informativeness to answer the question. Remarkably, all the procedures of SuRe are conducted via *zero-shot prompting*. Consequently, SuRe is widely applicable when LLMs are only accessible with black-box API, even without query-relevant few-shot examples.

Through the experiments on four different QA datasets, we demonstrate the effectiveness of SuRe for improving the zero-shot ODQA performance of retrieval-augmented LLMs. For example, we observe that the augmentation of 10 relevant passages effectively improves QA accuracy (up to 8.2% with Contriever (Izacard et al., 2022)) of ChatGPT (OpenAI, 2022), and the gain is significantly enlarged with SuRe (up to 12.8%), as shown in Figure 1. Overall, SuRe with ChatGPT and BM25 (Robertson et al., 2009) exhibited 4.6%/4.0% exact match (EM)/F1 score improvements compared to the standard prompting in average on four ODQA datasets. In addition, SuRe is well generalized to different configurations of various retrieval methods and LLMs. More interestingly, we observe that the generated summarization by SuRe could be further utilized to evaluate the importance of the retrieved passages, and also verify that it has a higher model/human preference as a rationale for the given prediction, compared to the generic summarization of retrieved passages. Overall, these results highlight the effectiveness of SuRe, to improve ODQA systems based on LLMs, not only in terms of accuracy but also of additional advantages that can improve the user experience. We, therefore, hope that the proposed framework could be beneficial in various real-world applications.

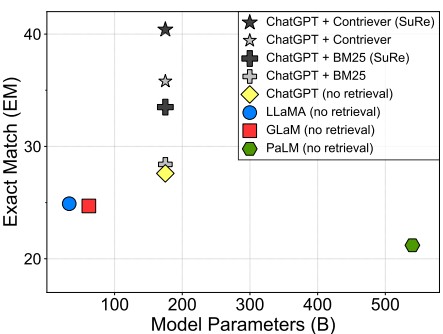

Figure 1: Zero-shot QA accuracy with various LLMs on Natural Question (Kwiatkowski et al., 2019). The performances of LLaMA-33B, GLaM-62B, and PaLM-540B are from the corresponding papers, respectively (Chowdhery et al., 2022; Du et al., 2022; Touvron et al., 2023a).

## 2 RELATED WORK

**Open-domain question answering.** Open-domain question answering (ODQA) (Voorhees et al., 1999) is a task that requires responding to factual questions using external knowledge sources (Zhu et al., 2015; Nagel, 2016). Recently, there has been significant research interest in ODQA systems, under a framework known as the *retriever-and-read* system (Chen et al., 2017). The role of *retriever* is to extract the relevant pieces of information from the given knowledge sources. For the retriever, there are two different popular methods: one is a lexical-based retriever, *e.g.*, TF-IDF or BM25 (Robertson et al., 2009), and the other is a sentence embedding-based retriever such as DPR (Karpukhin et al., 2020) or Contriver (Izacard et al., 2022). On the other hand, the *reader* is responsible for aggregating and reasoning with the retrieved information to generate answers. Usually, recent transformer-based language models (LMs) such as BERT (Kenton & Toutanova, 2019) or T5 (Raffel et al., 2020) are widely adopted for the reader after fine-tuning. In contrast, LLMs exhibit

comparable performance or outperform in QA without fine-tuning (Kamalloo et al., 2023; Shi et al., 2023), which indicates a potential to serve as a universal QA system (Xuan-Quy et al., 2023).

**Retrieval-augmented language models.** Similar to enhancing QA systems with retriever in ODQA, augmenting LMs with relevant information retrieved from external knowledge sources has been demonstrated as an effective way to improve the performance of LMs on various NLP tasks (Guu et al., 2020; Lazaridou et al., 2022; Min et al., 2022; Liu et al., 2023a), by reducing hallucination of LLMs and leveraging external knowledge which is not seen during pre-training. To construct such retrieval-augmented LMs, the standard approach is conducting additional fine-tuning to learn how to incorporate the retrieved information (Guu et al., 2020; Borgeaud et al., 2022; Izacard et al., 2023). However, when considering the recent nature of LLMs with increasing scale and providing black-box API only, such a direction becomes less attractive. One promising direction to address this challenge is investigating a better *prompting* (Brown et al., 2020), which incorporates the retrieved information as additional inputs in a sophisticated way. However, this direction has been only limitedly explored. Appending the retrieval (Si et al., 2023; Trivedi et al., 2023) is a common practice for prompting, but Liu et al. (2023b) recently revealed its limitation in utilizing the retrieved information. Aggregating the predictions from each retrieved passage has been also explored (Lazaridou et al., 2022; Shi et al., 2023), but LLMs can't see a full context of retrieved information in this case. More discussions about the summarization of retrieval in open-domain context are in Appendix G.

## 3 SUMMARIZED RETRIEVAL FOR QUESTION ANSWERING

### 3.1 OVERVIEW AND PROBLEM DESCRIPTION

**Overview.** In this section, we present our framework, coined Summarized Retrieval (SURE) to enhance ODQA performance of LLMs, by proposing an improved way to incorporate retrieved passages for the prediction. Our main idea is to construct multiple summaries of the retrieved passages conditioned with each of a few answer candidates, and predict the most plausible candidate as the answer after evaluating the validity and relative informativeness of summaries. In Sections 3.2 and 3.3, we present the details to generate the summarizations and evaluate them. Figure 2 presents the specific example of QA procedure via SURE.

**Problem description.** Open-domain question answering (ODQA) is an extension of QA tasks that answer questions that require background knowledge by leveraging an external database. In order to answer the given question $q$, the ODQA system typically follows *retrieve-and-read* framework (Chen et al., 2017; Lee et al., 2019), where the *retriever* finds the informative passages $C_N^+$ from the whole corpus $C$, and the *reader* exploits the retrieved passages to decide the answer $a$, which can be formulated as follows:

$$C_N^+ = \texttt{Retriever}(q, C, N) \text{ and } \widehat{a} = \texttt{Reader}(q, C_N^+), \tag{1}$$

where $N$ is the number of retrieved passages and $\widehat{a}$ is the predicted answer.

In this work, we focus on improving a prompting method for an LLM-based ODQA system. Specifically, we adopt the existing *retriever* method, *e.g.*, BM25 (Robertson et al., 2009) or Contriever (Izacard et al., 2022), with the dataset-specific corpus. For the *reader* method, we use LLMs, denoted by $\mathcal{M}$, such as ChatGPT (Sun et al., 2023) or LLaMA-2 (Touvron et al., 2023b), by incorporating the retrieved passages via *prompting* (Brown et al., 2020) without additional training. For example, with a prompt $p(q, C_N^+) =$ "Reading passages $C_N^+$, answer to question $q$", the prediction $\widehat{a}$ is obtained from $\mathcal{M}$, *i.e.*, $\widehat{a} = \mathcal{M}\big(p(q, C_N^+)\big)$.

### 3.2 CONDITIONAL SUMMARIZATION OF RETRIEVED PASSAGES

To better exploit the retrieved passages with LLMs, SURE first summarizes them conditioned on each of a few potential answer candidates. This *conditional* summarization of retrieved passages would include the specific contexts supporting a given answer candidate, compared to the generic summarization focusing on the wide coverage for the retrieved passages. Specifically, SURE first generates answer candidates and then conducts conditional summarization.

**Candidates generation.** Given a question $q$, retrieved passages $C_N^+$, and LLM $\mathcal{M}$, we first generate $K$ answer candidates $\widetilde{\mathbf{y}} = [\widetilde{y}_1, \ldots, \widetilde{y}_K]$ using a prompt $p_{\texttt{can}}$ designed for candidate generation from

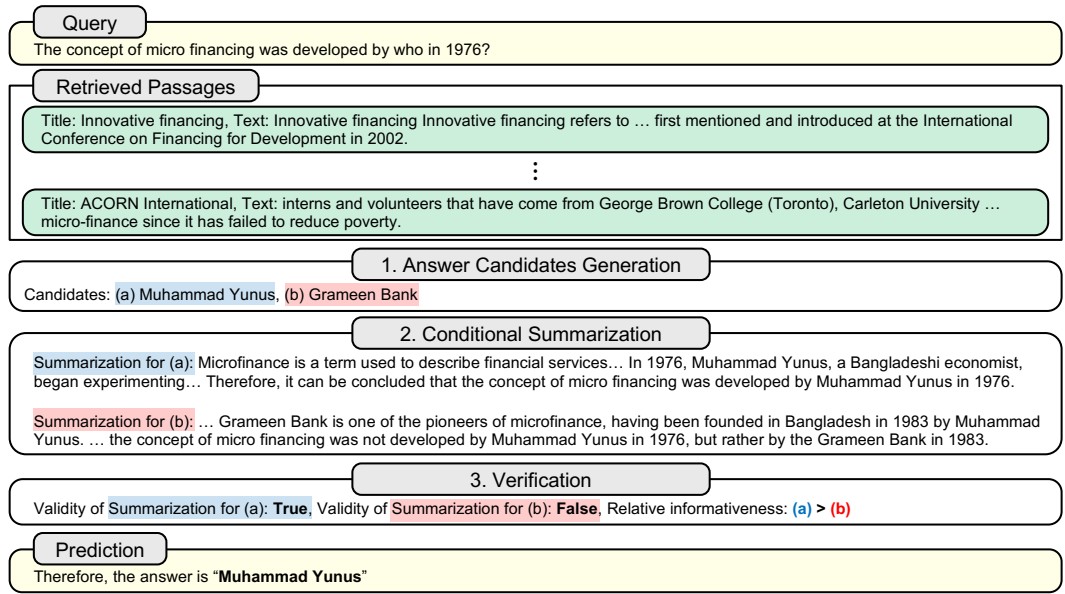

Figure 2: Example of QA with the proposed SURE framework. Given a query question and relevant passages retrieved by an external method, *e.g.*, BM25 (Robertson et al., 2009), a large language model, *e.g.*, ChatGPT, needs to predict the answer. To improve this, SURE first generates multiple answer candidates via prompting, and then conditionally summarizes the retrieved passages to support each candidate. By comparing the validity and relative informativeness of summaries, SURE selects the most plausible candidate as a final prediction.

$q$ and $C_N^+$:

$$\widetilde{\mathbf{y}} = \mathcal{M}\left(p_{\mathtt{can}}(q, C_N^+)\right). \tag{2}$$

In Figure 2, one can observe the example of generated candidates. It is noticeable that the previous works utilized stochastic decoding to generate multiple answer candidates (Lazaridou et al., 2022; Weng et al., 2022). However, we empirically observe that explicitly prompting an LLM to generate $K$ potential candidates outputs more diverse and high-quality candidates.

**Candidate-conditioned summarization.** Next, we *conditionally* summarize the retrieved passages $C_N^+$ focusing on including the relevant contexts to validate each candidate $\widetilde{y}_k \in \widetilde{\mathbf{y}}$ as an answer to $q$:

$$s_k = \mathcal{M}\left(p_{\mathtt{sum}}(q, C_N^+, y_k)\right) \text{ for } k = 1, \ldots, K \tag{3}$$

where $p_{\mathtt{sum}}$ is a prompt to obtain the conditional summarization $s_k$ from $q$, $C_N^+$, and $\widetilde{y}_k$. We present some examples of the generated summarizations in Figure 2, and more examples are in Appendix B. Remarkably, the generated summarizations effectively reduce the given passages by focusing on extracting the candidate-relevant contexts (*e.g.*, 1035 words of retrieved passages → 93 words of summarization). Also, we verify that the contexts of the generated summarization are specialized on a given answer candidate; when we measure TF-IDF (Chowdhury, 2010) based text similarity between two candidates and two conditional summarizations from each candidate (*e.g.*, summarization #1 is generated to support answer candidate #1) on Natural Question dataset (Kwiatkowski et al., 2019) in Figure 3, the summarization exhibits a higher similarity with the corresponding candidate than the other candidate.

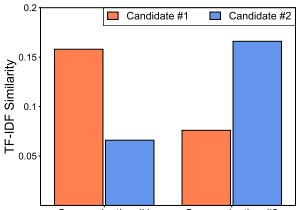

Figure 3: TF-IDF overlap between candidates and conditional summarizations.

### 3.3 SELECTIVE PREDICTION VIA VERIFICATION OF SUMMARIZATIONS

Then, using the generated summarizations, SURE confirms the most plausible answer among the candidate set for the prediction. Our key intuition is that the quality (*e.g.*, factuality, logicality, and

---
**Algorithm 1** SURE algorithm
---
1: **Input:** Large language model $\mathcal{M}$, question $q$, $N$ retrieved passages $C_N^+$, candidate number $K$
2: **Answer Candidate Generation:** $\widetilde{\mathbf{y}} = \mathcal{M}\left(p_{\text{can}}(q, C_N^+)\right)$, $\widetilde{\mathbf{y}} = [\widetilde{y}_1, \dots, \widetilde{y}_K]$
3: **Conditional Summarization:** $s_k = \mathcal{M}\left(p_{\text{sum}}(q, C_N^+, y_k)\right)$ for $k = 1, \dots, K$
4: **Instance-wise Validation:** $v(s_k) \leftarrow$ Eq. 4 with $\mathcal{M}\left(p_{\text{val}}(q, s_k)\right)$
5: **Pair-wise Ranking:** $r(s_k, S_K)$, $r_{\text{pair}}(s_k, s_i) \leftarrow$ Eq. 5 with $\mathcal{M}\left(p_{\text{rank}}(q, s_k, s_i)\right)$
6: **Output:** Prediction $\widehat{a} = \widetilde{y}_{k^*}$, $k^* = \arg\max_k v(s_k) + r(s_k, S_K)$
---

readability) of the generated summarizations would vary depending on the plausibility of answer candidates, so as more plausible the answer, the corresponding summarization also will be more plausible. Then, LLMs can find the most plausible summarization among these multiple summarizations if a proper evaluation way is given. To this end, we propose to evaluate the generated summarizations with *instance-wise* validity and *pair-wise* ranking among them.

**Instance-wise validity.** First, we evaluate the validity of each summarization $s_k$ whether it is not a degenerated case as the provided passages are not enough to support $\widetilde{y}_k$, or it properly supports the given answer candidate $\widetilde{y}_k$, rather than the other candidate $\widetilde{y}_i$, $i \neq k$.[2] To be specific, we measure a validity $v_k$ of each summarization $s_k$ using a prompt $p_{\text{val}}$ designed for the validation:

$$v(s_k) = 1, \text{ when } \mathcal{M}\left(p_{\text{val}}(q, y_k, s_k)\right) = \text{True} \quad \text{or} \quad v(s_k) = 0, \text{ else.} \tag{4}$$

**Pair-wise ranking.** In addition, we evaluate how the given summarization $s_k$ is *relatively informative* to answer the question $q$, among all summaries $S_K = \{s_k\}_{k=1}^K$. To this end, we measure a ranking $r_k$ using a pair-wise ranking prompts (Qin et al., 2023; Sun et al., 2023):

$$r(s_k, S_K) = \sum_{i \neq k}^K r_{\text{pair}}(s_k, s_i), \ r_{\text{pair}}(s_k, s_i) = \begin{cases} 1, & \mathcal{M}\left(p_{\text{rank}}(q, s_k, s_i)\right) = s_k \\ 0, & \mathcal{M}\left(p_{\text{rank}}(q, s_k, s_i)\right) = s_i \\ 0.5, & \text{else} \end{cases}, \tag{5}$$

where $p_{\text{rank}}$ is a prompt to determine which is relatively more informative one to answer the question by comparing two summaries. To prevent the order bias of LLMs (Zhao et al., 2021), we query the same pair of summaries twice by changing their order at the prompt $p_{\text{rank}}$.

Finally, SURE makes a final prediction $\widehat{a}$ by incorporating both $v(s_k)$ and $r(s_k, S_K)$:

$$\widehat{a} = \widetilde{y}_{k^*}, \ k^* = \arg\max_k v(s_k) + r(s_k, S_K), \tag{6}$$

*i.e.*, both validity and ranking scores are *equally* contributed. Algorithm 1 summarizes the formal procedure of SURE. We also highlight that the common prompts are shared across different datasets and LLMs, and the used prompts $p_{\text{can}}, p_{\text{sum}}, p_{\text{val}}, p_{\text{rank}}$ are presented in Appendix A.

## 4 EXPERIMENTS

In this section, we design our experiments to investigate the following questions:

○ Does SURE improve the accuracy of LLMs on various ODQA datasets? (Table 1)
○ Is SURE generalizable across various retrieval methods and LLMs? (Table 2)
○ What is the effect of each component in SURE? (Table 3)
○ Is SURE's summarization a good rationale for the answer? (Table 4 & Figure 4)

### 4.1 SETUPS

**Evaluation datasets.** For all experiments, we measure zero-shot QA accuracy with the four different ODQA datasets: (1) Natural Questions (NQ) (Kwiatkowski et al., 2019), (2) WebQuestions (WebQ) (Berant et al., 2013), (3) 2WikiMulti-hopQA (2Wiki) (Ho et al., 2020), and (4) HotpotQA (Yang et al., 2018). For NQ and WebQ, we use their original test splits and 21M English Wikipedia dump

---
[2] We present such failure cases in Appendix D.

Table 1: EM / F1 for different QA methods with ChatGPT on four QA datasets. $N = 10$ most relevant passages are retrieved using BM25, except *no retrieval*. The best and second best scores are highlighted in **bold** and underline, respectively.

| Methods / Datasets | NQ | WebQ | 2Wiki | HotpotQA | Average |
|---|---|---|---|---|---|
| No retrieval | 27.6 / 39.0 | 25.0 / **38.8** | 21.4 / 24.8 | 22.2 / 31.9 | 24.1 / 33.6 |
| Base | 28.4 / 38.8 | 19.6 / 32.5 | 27.4 / 32.8 | 30.8 / 40.3 | 26.6 / 36.1 |
| Rerank | 24.8 / 33.9 | 18.8 / 30.6 | 23.0 / 28.4 | 27.8 / 37.4 | 23.6 / 32.6 |
| RePlug | 26.0 / 35.3 | 18.8 / 31.5 | 23.6 / 28.5 | 28.0 / 37.9 | 24.1 / 33.3 |
| Selection-inference | 24.3 / 32.8 | 17.3 / 28.6 | 22.6 / 29.5 | 30.8 / 39.6 | 23.8 / 32.6 |
| Chain-of-thoughts | 22.3 / 31.4 | 15.2 / 27.8 | 19.6 / 22.5 | 25.6 / 31.8 | 20.7 / 28.4 |
| Self-verification | 25.2 / 35.4 | 16.1 / 28.5 | 23.2 / 30.5 | 31.6 / 41.8 | 24.0 / 34.1 |
| SURE (Ours) | **33.5 / 42.3** | **25.1** / 36.6 | **32.8 / 38.1** | **33.2 / 43.4** | **31.2 / 40.1** |

(Karpukhin et al., 2020) as the source passages for the retrieval. For 2Wiki and HotpotQA, we use the subsampled splits released by Trivedi et al. (2023), along with the corresponding corpus for each data. For the experiments with LLaMA2-chat (Table 2) and more analyses (Section 4.3), we took 500 randomly subsampled examples of NQ and WebQ datasets for efficient experiments considering limited computing resources, and denoted these datasets NQ* and WebQ*, respectively. As evaluation metrics, we calculate the exact match (EM) and F1 score. The EM accuracy is the ratio of correct answers in the test dataset, where a given prediction is considered correct if it coincides with one of the gold answers. The F1 score measures the overlap between bags of tokens in the prediction and the gold answer. We normalize the predictions and answers (*i.e.*, case-folded, and punctuation) to compute the metrics, following the implementation of Rajpurkar et al. (2016).

**Baselines.** We compare SURE with the following baselines. (1) *No retrieval* answers the question with LLMs without the retrieved passages (*i.e.*, closed-book setup). (2) *Base* appends the retrieved passages as additional inputs of LLMs via prompting. (3) Line of works for better exploitation of retrieved passages with LLMs: *Rerank* (Lazaridou et al., 2022) and *RePlug* adopt an ensemble strategy that makes predictions based on each passage and then aggregates them with specific voting methods. Specifically, *Rerank* and *RePlug* utilize TF-IDF and sentence embedding from Contriever, respectively. (4) Adapt the works that incorporate intermediate reasoning steps for improved reasoning with LLMs, as summarizing could be viewed as a specific type of reasoning: *Selection-inference* (Creswell et al., 2023) measures the ranking of the passages, and conducts interactive answering by adding the passages one by one starting from higher ranked ones. *Chain-of-thoughts* (Kojima et al., 2022): we add zero-shot Chain-of-thoughts prompting (Wei et al., 2022) into the prompt of *Base*. *Self-verification* (Weng et al., 2022) generates answer candidates based on random sampling, then selects the most plausible one by verifying its reasoning with the question from conditional masking.

**Implementation details.** For the experiments, we use three recent state-of-the-art LLMs: Chat-GPT (`gpt-3.5-turbo-0301`) (OpenAI, 2022), GPT-4 (`gpt-4-0613`) (OpenAI, 2023), and LLaMA2-chat-70B (Touvron et al., 2023b). We use a temperature of 0.0 when calling the API or greedy decoding for LLaMA, to remove the effect of random sampling (Sun et al., 2023). For the retrieval methods, we use three different approaches: BM25 (Robertson et al., 2009), DPR-multi (DPR) (Karpukhin et al., 2020), and Contriever (Izacard et al., 2022). We use the implementations in Elasticsearch for BM25, and BEIR for DPR and Contriever, respectively.[3] In the case of SURE, we use the same prompts across the different datasets, and they are presented in Appendix A. Also, we use a fixed value of $K = 2$ during the experiments since we observe that the improvements by increasing $K$ are limited, as shown in Appendix B. When there are multiple candidates with equal plausibility (Eq. 6), then SURE selects the one generated earlier in Eq. 2.

## 4.2 MAIN RESULTS

Table 1 summarizes the experimental results on four different ODQA datasets, under ChatGPT with $N = 10$ retrieved passages using BM25. First, augmenting the retrieved passages with prompting is effective in improving ODQA accuracies of LLMs. For example, the average EM across four ODQA datasets is increased from 24.1 to 26.6. Somewhat surprisingly, we observe that *Base* outperforms

---

[3] https://www.elastic.co/, https://github.com/beir-cellar/beir

Table 2: EM with different configurations of LLMs and retrieval methods on four QA datasets. $N = 10$ most relevant passages are commonly retrieved. F1 scores are reported in Table 5. For LLaMA2-chat, we conducted experiments on NQ* and WebQ* and the results are indicated by *.

| Datasets | ChatGPT | | | | | | GPT-4 | | LLaMA2-chat | |
|---|---|---|---|---|---|---|---|---|---|---|
| | BM25 | + SURE | DPR | + SURE | Contriever | + SURE | BM25 | + SURE | BM25 | + SURE |
| NQ | 28.4 | **33.5** | 36.1 | **41.0** | 35.8 | **40.4** | 30.2 | **32.4** | 18.6* | **30.4*** |
| WebQ | 19.6 | **25.1** | 23.2 | **27.3** | 22.5 | **28.7** | 21.5 | **21.7** | 16.0* | **24.0*** |
| 2Wiki | 27.4 | **32.8** | 19.2 | **21.4** | 27.2 | **32.6** | 34.8 | **38.2** | 20.2 | **27.8** |
| HotpotQA | 30.8 | **33.2** | 25.6 | **27.4** | 32.2 | **33.6** | 34.8 | **40.6** | 24.0 | **28.0** |
| Average | 26.6 | **31.2** | 26.0 | **29.3** | 29.4 | **33.8** | 30.3 | **33.2** | 19.7 | **27.6** |

Table 3: Ablation and more analyses. EM / F1 with ChatGPT are compared on four QA datasets. $N = 10$ most relevant passages are retrieved using BM25. The best scores are highlighted in **bold**.

| Methods / Datasets | NQ* | WebQ* | 2Wiki | HotpotQA | Average |
|---|---|---|---|---|---|
| Base | 29.4 / 41.7 | 19.4 / 32.2 | 27.4 / 32.8 | 30.8 / 40.3 | 26.8 / 36.8 |
| Conditional summarizations | 30.4 / 40.9 | 20.8 / 33.5 | 29.2 / 34.5 | 33.0 / **43.4** | 28.4 / 38.1 |
| + Pair-wise ranking | 30.6 / 41.2 | 21.6 / 34.8 | 31.0 / 36.0 | 30.6 / 40.7 | 28.5 / 38.2 |
| + Instance-wise validity (SURE) | **35.6** / 44.9 | **23.2 / 36.5** | **32.8 / 38.1** | 33.2 / **43.4** | **31.2 / 40.7** |
| MCQ prompt | 35.2 / **45.3** | 22.4 / 35.1 | 30.4 / 36.1 | 31.0 / 41.5 | 29.8 / 39.5 |
| Sum-and-pred (Gen) | 26.4 / 37.8 | 19.8 / 32.6 | 25.6 / 32.3 | **33.8** / 43.3 | 27.3 / 37.1 |

other sophisticated baselines overall; this inefficiency of previous methods might be a result of a more challenging yet practical experimental setup. For example, we assume the zero-shot QA rather than few-shot setups, and also consider general black-box APIs for LLMs which do not provide the output probability. In contrast, one can observe that SURE successfully improves QA accuracy of LLMs by effectively exploiting the retrieved passages. In particular, SURE exhibits 4.6%/4.0% absolute EM/F1 improvements in the average, compared to naïvely appending the retrieved passages.

We further demonstrate the compatibility of SURE across various LLMs and retrieval methods. Specifically, in addition to ChatGPT and BM25 considered in Table 1, we run experiments on three different LLMs (GPT-4, and LLaMA2-chat) and two different retrieval methods (DPR and Contriever). In Table 2, we compare EM metric of SURE with the baseline that simply appends the retrieved passages. Here, ODQA performance significantly depends on the retrieval methods and types of LLMs; for example, using Contriever instead of BM25 makes 2.8% average EM improvements, and using GPT-4 instead of ChatGPT makes 3.7% average EM improvements, respectively. Overall, one can observe that SURE consistently improves ODQA accuracy regardless of types of LLMs and retrieval methods, with 4.6% average EM improvements. More interestingly, SURE successfully improves average EM scores of LLaMA2-chat as 7.9%, a state-of-the-art open-sourced LLM, which further indicates the practical usefulness of SURE as a simple yet effective solution for ODQA for the open source research community. The F1 results are presented in Appendix B.1.

## 4.3 ADDITIONAL ANALYSES

In this section, we conduct additional analyses of SURE. We conduct experiments using ChatGPT as an LLM, BM25 as a retriever, NQ* and WebQ* as datasets.

**Ablation and more analysis of SURE.** First, we compare the following methods for the ablation of SURE: (1) *Base*: appends the retrieved passages to inputs, (2) + *Conditional summarizations*: additionally appends all the conditional summarizations, (3) + *Pair-wise ranking*: selects the summarization with only ranking (Eq. 4), and (4) + *Instance-wise validity*: selects the summarization with both ranking and validity, *i.e.*, SURE. In addition, we consider two different methods to further analyze where the effectiveness of SURE comes from. (5) *MCQ prompt*: composes Multiple Choice Questions by generating the answer candidates via prompting (Eq. 2) and using them as possible choices for prediction by appending them to input prompt (Robinson et al., 2023) (more details in Appendix A.7), (6) *Sum-and-pred (Gen)*: instead of *conditional* summarization, it generates generic summarization and predicts the answer based on it. We present the results in Table 3.

Table 4: Comparison as reranking method. EM / F1 with ChatGPT are compared on four QA datasets. A single most relevant passage is selected among $N = 10$ passages retrieved by BM25. The best scores are highlighted in **bold**.

| Datasets / Methods | NQ* | WebQ* | 2Wiki | HotpotQA | Average |
|---|---|---|---|---|---|
| BM25 | 12.6 / 18.8 | 9.0 / 17.6 | 14.8 / 18.1 | 21.8 / 28.3 | 14.6 / 20.7 |
| Sent-encoder (q) | 18.2 / 26.4 | 11.4 / 21.1 | 14.8 / 18.1 | 20.2 / 27.2 | 16.2 / 23.2 |
| LLM-rerank | 20.0 / 28.4 | 14.2 / 24.4 | **18.2** / 21.3 | 26.0 / 34.4 | 19.6 / 27.1 |
| Sent-encoder (Gen) | 21.2 / 31.7 | 13.6 / 25.3 | 17.8 / 21.1 | 27.0 / 34.3 | 19.9 / 28.1 |
| Sent-encoder (SuRE) | **23.2 / 32.5** | **15.4 / 28.0** | 18.0 / **21.5** | **28.8 / 36.7** | **21.4 / 29.7** |

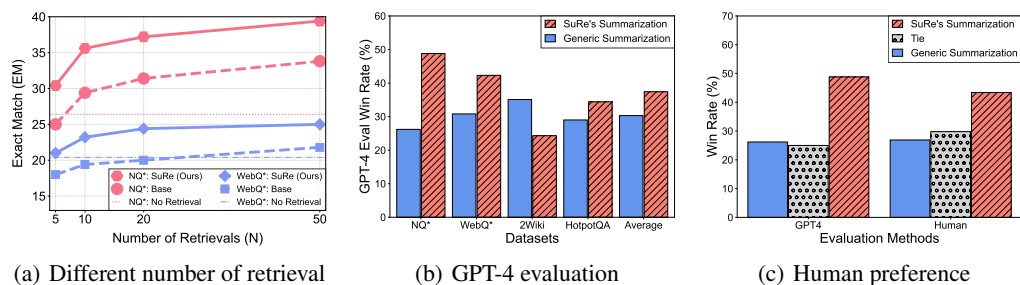

(a) Different number of retrieval     (b) GPT-4 evaluation     (c) Human preference

Figure 4: (a) EM with different numbers of retrieved passages ($N$) under ChatGPT and BM25. (b) Comparison between SuRE's summarization and generic summarization via GPT-4 evaluation (Liu et al., 2023c). (c) Human preference between SuRE's summarization and generic summarization on 84 samples of NQ*, along with GPT-4 evaluation. More results are in Appendix C.

First, constructing conditional summarizations improves performance as they can extract specialized contexts for a given question and its answer candidates. Next, incorporating the evaluation on the instance-wise validity of each summarization significantly improves the performance compared to only considering the ranking among summarizations, as it enables more precise selection by adding the assessment regarding the relevance and coherence of the summarization in relation to the given question and prediction pair. Also, a simple aggregation of generated answer candidates in the prompt shows improvement, which indicates the effectiveness of our generated candidates. However, this method becomes inefficient when the given question requires more complex reasoning to answer. Lastly, using generic summarization is effective in improving ODQA with LLMs by providing concentrated and brief contexts and addressing the difficulty from the long context (Liu et al., 2023b). However, the gain is significantly limited compared to SuRE, which demonstrates that the key components of SuRE are conditional summarization and comparison, rather than simply providing compressed contexts.

**Different number of retrieval.** Next, we investigate the effect of the number of retrieved passages ($N$). Increasing $N$ is one of the most intuitive ways to improve the performance of *retrieve-and-read* system by providing more extensive information (Karpukhin et al., 2020), and hence it is natural to expect that similar positive results could be observed with retrieval-augmented LLMs. However, on the other hand, its effectiveness could be limited as LLMs could fail to handle long input contexts (Liu et al., 2023b). To verify the effect of different $N$ on retrieval-augmented LLMs using prompting, we measure EM of ChatGPT and BM25 with varied $N$. In Figure 4(a), we present the results of *Base* and SuRE on NQ* and WebQ*. First, we observe that the accuracy of retrieval-augmented LLMs significantly depends on $N$; when a small number of retrieved passages is only available, the performance of *Base* could be even behind the performance without retrieval, as it restricts the prediction within the limited contexts. As $N$ increases, its performance is increased and takes benefit from the retrieval system. With SuRE, the accuracy of LLMs could be improved even with the small number of retrievals ($N = 5$), and it achieves better accuracy with larger $N$.

**Effectiveness for finding important passages.** In previous experiments, we mainly focus on demonstrating the effectiveness of SuRE for improving QA accuracy. While the accurate answer is the most important feature of the QA system, providing the proper rationale for the answer is another important feature, especially in LLM-based systems for reliable usage by users such as search engines. One of the standard approaches for this is explicitly enumerating the most relevant

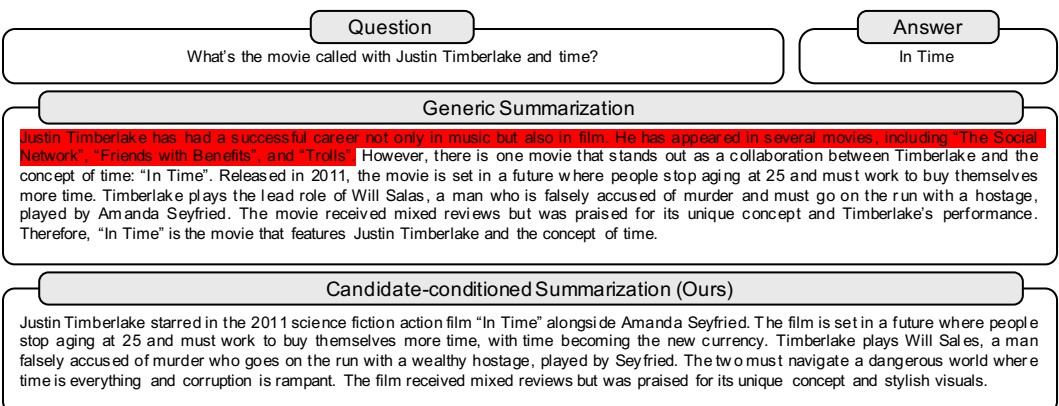

Figure 5: Qualitative comparison of candidate-conditioned summarization from SURE (Ours) compared to generic summarization as a rationale for the answer. More examples are in Appendix B.

retrieved passages based on the specific scoring method, which is often called *Re-ranking* (Nguyen et al., 2016; Izacard et al., 2023). To explore the advantages of SURE in this aspect, we measure QA accuracy of ChatGPT augmented with the one passage considered to be most relevant with a specific reranking method within $N = 10$ originally retrieved passages with BM25. To extract such a reranking method for SURE, we use the cosine similarity between the sentence embeddings (Reimers & Gurevych, 2019) of the generated summarization and the retrieved passages, denoted by *Sent-encoder (*SURE*)*. Then, we compare it with the following baselines for reranking: (1) *BM25*: original retrieval score, *i.e.*, no reranking, (2) *Sent-encoder (q)*: sentence-encoder-based reranking using the similarity between retrieved passages and question (Nguyen et al., 2016), (3) *LLM-rerank*: LLM-based reranking (Sun et al., 2023), and (4) *Sent-encoder (Gen)*: sentence-encoder-based reranking using the similarity between retrieved passages and generic summarization. The results are presented in Table 4. Here, we observe that all the reranking methods are effective compared to no reranking. In addition, LLM-based reranking shows a higher accuracy, while SURE's similarity-based reranking outperforms all the baselines, demonstrating the superiority of SURE.

**Qualitative evaluation as rationale to answer.** Lastly, we explore the additional benefits of SURE, which offers rationales to support the prediction. Specifically, we compare the summarization from SURE with the generic summarization, which is also generated by LLMs but with no constraint of supporting specific answer candidates. To separately consider the quality as rationale with the accuracy of prediction, we only compare the samples correctly predicted by both SURE and *Generic summarization* used in Table 3; for example, it results in 84 remaining samples in the case of NQ*. We first evaluate using GPT-4, which has been demonstrated to have a high correlation with humans (Liu et al., 2023c). We present the results in Figure 4(b). Here, one can observe that the summarization via SURE is more preferred by GPT-4; for example, Generic summarization wins 30.3% while SURE wins 37.4% on average. It is also worth noting that the average length of both summarizations is similar (Generic: 600 vs SURE's: 570 average characters on NQ), therefore the bias of GPT to prefer the longer response (Wang et al., 2023) might limitedly affect the result. Next, we ask human evaluators *which summarization is more informative and plausible to support the given question-answer pair* on 84 samples of NQ*. This result is presented in Figure 4(c). Here, we also observe a higher preference for SURE's summarization (Generic: 26.9% vs SURE: 43.4%). Overall, these results reveal the potential of SURE toward a better ODQA system by providing a high-quality rationale for the answer. Details on human evaluations are presented in Appendix C.

## 5 CONCLUSION

In this paper, we proposed SURE, a simple yet effective framework to improve ODQA accuracy of LLMs. Our key idea is to ensure the correctness of predicted answers by constructing the summaries of the retrieved passages for the potential answer candidates and evaluating their validity and ranking. Our experiments demonstrate that SURE significantly improves ODQA performance of various retrieval-augmented LLMs, and also has additional advantages for measuring the importance of passages and providing the rationale for prediction. From these advantages, we believe our framework can contribute to various real-world applications and provide a better experience to users.

## ETHICS STATEMENT

We strongly believe that SURE can provide a strong positive impact in real-world applications related to QA, *e.g.*, search engines or chatbots. Since SURE can provide the summarization that supports the corresponding prediction specifically, it can significantly improve the explainability (Mao et al., 2022) and reliability (Whitehead et al., 2022) of QA systems which are more important when they are constructed using black-box LLMs. Moreover, considering the success of LLMs in various applications more than QA (Izacard et al., 2023; Nam et al., 2023), we expect the advantages of this framework to better exploit the retrieved passages with LLMs will be beneficial to them.

In contrast, there also exists some potential negative impacts when developing a system with the multiple usages of LLMs, as it could be costly (Chen et al., 2023) and generate sensitive (Santurkar et al., 2023) and malicious (Deshpande et al., 2023) text outputs. Since the summarization from SURE is constructed based on the provided passages, one should consider their quality to prevent undesirable outputs. On the other hand, incorporating the additional filtering could be a strong solution (Le Bras et al., 2020; Schick et al., 2021). To reduce the cost, substituting specific steps of SURE, *e.g.*, measuring validity, with trainable small LMs could be an effective way, similar to Yang et al. (2020); Lewis et al. (2021); Li et al. (2023).

## REPRODUCIBILITY STATEMENT

We provide implementation details (*e.g.*, design of prompts, used APIs, and retrieval methods) and experiment setups (*e.g.*, datasets and metrics) in Section 4 and Appendix A. In addition, we will release source codes near future.

## ACKNOWLEDGMENTS

This work was mainly supported by Institute of Information & communications Technology Planning & Evaluation (IITP) grant funded by the Korea government (MSIT) (No.2021-0-02068, Artificial Intelligence Innovation Hub; No.2019-0-00075, Artificial Intelligence Graduate School Program (KAIST)). This work was partly supported by KAIST-NAVER Hypercreative AI Center.

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

## A   DESIGNED PROMPTS FOR EXPERIMENTS

In this section, we present the specific prompts used for the experiments in Section 4.

### A.1   ANSWER CANDIDATES GENERATION

In Listing 1, we present the prompt $p_{\text{can}}$ which is used to generate $K$ answer candidates $\widetilde{\mathbf{y}} = [\widetilde{y}_1, \ldots, \widetilde{y}_K]$ from the given question and $N$ retrieved passages (Eq. 2). Here, we present the case of $K = 2$.

**Listing 1** Prompt for answer candidates generation.

```
f'''
Below are N passages related to the question at the end. After reading
↪  the passages, provide two correct candidates for the answer to the
↪  question at the end. Each answer should be in the form: (a) xx, (b)
↪  yy, and should not exceed 3 words for each candidate.

Passage #1 Title: {Passage #1 Title}
Passage #1 Text: {Passage #1 Text}

...

Passage #N Title: {Passage #N Title}
Passage #N Text: {Passage #N Text}

Question: {Question}

Answer:
'''
```

## A.2 CONDITIONAL SUMMARIZATION

In Listing 2, we present the prompt $p_{\texttt{sum}}$ which is used to generate conditional summarization $s_k$ of retrieved passages to validate each candidate $\widetilde{y}_k$ as an answer to the question (Eq. 3).

**Listing 2** Prompt for conditional summarization.

```
f'''
Passage #1 Title: {Passage #1 Title}
Passage #1 Text: {Passage #1 Text}

...

Passage #N Title: {Passage #N Title}
Passage #N Text: {Passage #N Text}

Your job is to act as a professional writer. You will write a
↪   good-quality passage that can support the given prediction about the
↪   question only based on the information in the provided supporting
↪   passages.

Now, let's start. After you write, please write [DONE] to indicate you
↪   are done. Do not write a prefix (e.g., "Response:") while writing a
↪   passage.

Question: {Question}
Choices: {(a) Choice 1 (b) Choice 2}
Prediction: {(a) Choice 1 (or (b) Choice 2)}
Passage:
'''
```

## A.3 INSTANCE-WISE VALIDATION

In Listing 3, we present the prompt $p_{\texttt{val}}$ which is used to evaluate the validity of each summarization $s_k$ whether it is not a degenerated case as the provided passages are not enough to support $\widetilde{y}_k$, or it properly supports the given answer candidate $\widetilde{y}_k$, rather than the other candidate $\widetilde{y}_i$, $i \neq k$ (Eq. 4).

**Listing 3** Prompt for instance-wise validation.

```
f'''
Question: {Question}

Prediction: {Prediction}

Passage: {Passage}

Does the passage correctly support the prediction? Choices: [True,
↪   False]. Answer:
'''
```

## A.4 PAIR-WISE RANKING

In Listing 4, we present the prompt $p_{\texttt{rank}}$ which is used to evaluate how the given summarization $s_k$ is *relatively informative* to answer the question $q$, among all summaries $S_K = \{s_k\}_{k=1}^K$ (Eq. 5).

**Listing 4** Prompt for pair-wise ranking.

```
f'''
Question: Given the following passages, determine which one provides a
↪   more informative answer to the subsequent question.

Passage 1: {Passage 1}

Passage 2: {Passage 2}

Target Question: {Question}

Your Task:
Identify which passage (Passage 1 or Passage 2) is more relevant and
↪   informative to answer the question at hand. Choices: [Passage 1,
↪   Passage 2].

Answer:
'''
```

## A.5 BASELINE PREDICTION

In Listing 5, we present the prompt that is used to append the retrieved passages of the question to give it as inputs of LLMs. The result with this prompt is denoted by *Base*, in Section 4. The same prompt is used for *no retrieval* by assuming $N = 0$, *i.e.*, the only question is given to LLMs with instruction.

**Listing 5** Prompt for baseline prediction.

```
f'''
Passage #1 Title: {Passage #1 Title}
Passage #1 Text: {Passage #1 Text}

...

Passage #N Title: {Passage #N Title}
Passage #N Text: {Passage #N Text}

Task description: predict the answer to the following question. Do not
↪  exceed 3 words.

Question: {Question}

Answer:
'''
```

## A.6 PROMPTS FOR GENERAL SUMMARIZATION

In Listing 6, we present the prompt that is used to construct *generic* summarization used in Section 4.3. One can observe that the conditioning part is removed, compared to $p_{\mathrm{sum}}$.

**Listing 6** Prompt for generic summarization.

```
f'''
Passage #1 Title: {Passage #1 Title}
Passage #1 Text: {Passage #1 Text}

...

Passage #N Title: {Passage #N Title}
Passage #N Text: {Passage #N Text}

Your job is to act as a professional writer. You will write a
↪  good-quality passage that can support the prediction about the
↪  question only based on the information in the provided supporting
↪  passages.

Now, let's start. After you write, please write [DONE] to indicate you
↪  are done. Do not write a prefix (e.g., "Response:") while writing a
↪  passage.

Question: {Question}
Passage:
'''
```

## A.7 PROMPTS FOR MCQ PROMPT

Recently, Robinson et al. (2023) demonstrated that multiple-choice prompts generally elicit much more accurate responses than do cloze prompts, for LLMs with high multiple-choice symbol bind-

ing ability like OpenAI Codex (Chen et al., 2021). Motivated by this, we consider MCQ prompt in Listing 7 and use it in Table 3, to evaluate the effectiveness of selecting the answer from the construction and verification of the conditional summarizations rather than direct prompting, under the same answer candidates from Eq. 2. One can observe that the conditioning with multiple choices part is added, compared to baseline prompting in Listing 5.

**Listing 7** Prompt for MCQ prompt.

```
f'''
Passage #1 Title: {Passage #1 Title}
Passage #1 Text: {Passage #1 Text}

...

Passage #N Title: {Passage #N Title}
Passage #N Text: {Passage #N Text}

Task description: predict the answer to the following question. Do not
↪   exceed 3 words.

Question: {Question}
Choices: {(a) Choice 1 (b) Choice 2}
Answer:
'''
```

### A.8 DESIGN PRINCIPLES FOR PROMPT

Before finalizing the prompts used in the experiments, we examined several prompt designs and chose the best-performing one. Here, we'd like to share two key observations from this process. First, precise and detailed instructions are crucial. As each component of the proposed framework operates in a zero-shot manner, its output greatly relies on the provided instruction. For example, in answer candidate generation (Eq. 2), the current prompt, outlined in Listing 1, consistently outperforms the initially considered simple prompt (`Task description: give two candidates for the answer to the following question (e.g., (a) xx, (b) yy)`). Second, proper input arguments are essential. For instance, along with the target candidate, providing all candidates as additional input enhanced the quality of conditional summarization. This is because it further specifies which contexts of retrieval should be the focus. However, including this information, or even the retrieval passages, disrupted the verification step by interrupting the focus on the summarizations.

## B ADDITIONAL QUANTITATIVE RESULTS

### B.1 MORE RESULTS FOR SURE UNDER DIFFERENT CONFIGURATIONS

In Table 5, we present F1 scores with different configurations of various LLMs and retrieval methods. Similar to the result in Table 2, it is observed that SURE consistently improves ODQA accuracy regardless of types of LLMs and retrieval methods, with 3.2% average F1 improvement on average.

### B.2 LIMITED ACHIEVABLE IMPROVEMENT WITH MORE CANDIDATES

As we denoted in Section 4.1, we use a fixed value of $K = 2$ for all the experiments. This is due to our initial observation that the room for improvement by increasing $K$ is not large compared to the additional costs. To investigate this, we first assume the method, denoted *Oracle*, which takes the maximum of EM and F1 among the multiple candidates, *e.g.,*, if one candidate is true and the other is wrong, then *Oracle* consider it as true. As one can see in Table 6, increasing $K = 3$ from $K = 2$ limitedly improves the accuracy (*e.g.*, 0.9% in EM), compared to the remaining room for improvement by better selection with small $K$; for example, there is 9.0% gap between SURE and *Oracle*, in terms of EM. Therefore, in this work, we keep $K = 2$ but we remark that SURE can be

Table 5: F1 with different configurations of LLMs and retrieval methods on four QA datasets. $N = 10$ most relevant passages are commonly retrieved. For LLaMA2-chat, we conducted experiments on NQ* and WebQ* and the results are indicated by *.

| Datasets | ChatGPT | | | | | | GPT-4 | | LLaMA2-chat | |
| --- | --- | --- | --- | --- | --- | --- | --- | --- | --- | --- |
| | BM25 | + SURE | DPR | + SURE | Contriever | + SURE | BM25 | + SURE | BM25 | + SURE |
| NQ | 38.8 | **42.3** | 47.4 | **50.8** | 47.6 | **50.4** | 40.9 | **42.4** | 36.4* | **42.9*** |
| WebQ | 32.6 | **36.6** | 37.5 | **40.4** | 37.7 | **40.9** | **36.4** | 32.1 | 35.3* | **40.5*** |
| 2Wiki | 32.8 | **38.1** | 22.8 | **25.2** | 32.9 | **37.1** | 39.2 | **43.2** | 31.2 | **36.2** |
| HotpotQA | 40.3 | **43.4** | 34.6 | **35.5** | 42.9 | **43.5** | 44.4 | **50.4** | **39.6** | 38.4 |
| Average | 36.1 | **40.1** | 35.6 | **38.0** | 40.3 | **43.0** | 40.2 | **42.0** | 35.6 | **39.5** |

Table 6: EM / F1 with different $K$ under ChatGPT. $N = 10$ most relevant passages are commonly retrieved with BM25.

| Datasets / Methods | NQ* | WebQ* | 2Wiki | HotpotQA | Average |
| --- | --- | --- | --- | --- | --- |
| No retrieval | 26.4 / 37.9 | 20.4 / 36.7 | 21.4 / 24.8 | 23.2 / 34.8 | 22.9 / 33.6 |
| Base | 29.4 / 41.7 | 19.4 / 32.2 | 27.4 / 32.8 | 30.8 / 40.3 | 27.0 / 36.9 |
| Oracle ($K = 2$) | 43.0 / 53.9 | 29.0 / 43.9 | 47.6 / 54.4 | 41.2 / 52.7 | 40.2 / 51.2 |
| Oracle ($K = 3$) | 45.2 / 56.0 | 29.8 / 47.2 | 48.4 / 56.2 | 41.0 / 54.0 | 41.1 / 53.4 |

extended with $K > 2$. Also, as there is remaining room for improvement, we hope that future work could reduce such a gap.

## B.3 ADDITIONAL EVALUATION WITH LLMS

In Section 4, we considered EM/F1 scores as the common metrics for the considered ODQA datasets, following the previous works (Chowdhery et al., 2022; Touvron et al., 2023a; Izacard et al., 2023; Shi et al., 2023), to make it easy to notice the significance of our results. Nevertheless, other factors like response coherence, relevance, and efficiency are important metrics to be considered.

To evaluate these aspects, we have conducted additional evaluations with LLMs approaches. Specifically, we measured two additional metrics: (1) MAUVE (Pillutla et al., 2021) and (2) LLM-acc (Kamalloo et al., 2023). MAUVE is a recently proposed metric to compare the two distributions of the text generation model and human-written text using divergence frontiers. MAUVE (scale of 0 to 100, higher is better) is known for correlating highly with human judgments, and is frequently used to evaluate LMs' responses (Su et al., 2022b; Gao et al., 2023). LLM-acc assesses the accuracy (%) of LLMs' responses to questions, using the prompting of LLMs instead of term overlap like EM/F1. We used the official code from the authors, only changing LLMs to ChatGPT. We measured this metric on NQ*, WebQ*, 2Wiki, and HotpotQA datasets, and the results are presented in Table 7.

Here, it is observed that the proposed method also makes significant improvements compared to the baseline under these two additional evaluations with LLMs approaches. Along with the results in Section 4, this result further validates that our framework enables LLMs to provide better answers to the given question.

## B.4 EXPERIMENTS ON LONG-FORM QUESTION ANSWERING

While we mainly conduct the experiments with QA datasets that have short answers in Section 4, our approach has the potential to be applicable beyond short-answer datasets. To verify this, we have conducted additional experiments on long-form question answering tasks to validate our approach's applicability. Specifically, we used ASQA dataset (Stelmakh et al., 2022; Gao et al., 2023) which consists of factoid questions and the corresponding long-form answers; for example, the answers of ASQA dataset have an average length of 71.8 words, while the answers of NQ dataset have 2.6 words. Following the setups in Gao et al. (2023), we compared the base prompting method with retrieval and name (ours) on 948 test examples, using ChatGPT (GPT-3.5-turbo-0301) with 5 retrieved passages via GTR (Ni et al., 2022) for the experiments. For the evaluation, we measure

Table 7: MAUVE (Pillutla et al., 2021) and LLM-evaluated accuracy (Kamalloo et al., 2023). We use ChatGPT and $N = 10$ most relevant passages are commonly retrieved with BM25.

| MAUVE / LLM-acc | NQ* | WebQ* | 2Wiki | HotpotQA | Average |
|---|---|---|---|---|---|
| Base | 81.3 / 53.2 | 61.3 / 48.8 | 35.1 / 36.2 | 62.4 / 51.6 | 60.0 / 47.5 |
| SURE (Ours) | 95.9 / 56.2 | 75.7 / 51.4 | 52.2 / 48.2 | 89.6 / 52.4 | 78.3 / 52.1 |

Table 8: Evaluation on ASQA dataset (Stelmakh et al., 2022). We use ChatGPT and $N = 5$ most relevant passages are commonly retrieved with GTR (Ni et al., 2022), following Gao et al. (2023).

| Methods / Metrics | ROUGE-L | STR-EM | MAUVE |
|---|---|---|---|
| Base | 38.00 | 39.81 | 69.83 |
| SURE (Ours) | 39.83 | 42.63 | 70.33 |

ROUGE-L and String Exact Match (STR-EM) for correctness, and MAUVE (Pillutla et al., 2021) for fluency and coherence, following the previous works (Stelmakh et al., 2022; Gao et al., 2023).

The results are presented in Table 8. One can observe that our proposed framework consistently improves the performance of retrieval-augmented LLMs for long-form QA tasks. However, we acknowledge that there is still room for improvement, particularly in finding better prompt designs, given that our current designs are based on performance on short-answer datasets. We hope future research will explore this direction, extending the benefits of our framework to broader QA scenarios with LLMs.

### B.5 EXPERIMENTAL WITH FEW-SHOT EXAMPLES

Here, we conduct additional experiments on NQ* and WebQ*, using 1-shot and 5-shot examples from training datasets during prediction. We compare the average EM/F1 of base prompting with retrieval and SURE, across four different random seeds used for sample selection. In Listing 8, we present the prompt that is used to generate $K$ answer candidates in the case where few-shot examples are given. Here, we present the case of $K = 2$. Note that if few-shot examples are provided, only the prompt for generating answer candidates is modified. Also, in Listing 9, we present the prompt for the base prompting. Table 9 shows that adding few-shot examples improves QA accuracy for both the baseline and name. Specifically, we observed that name's gain primarily results from generating more accurate answer candidates. These findings suggest that our proposed method could be effective in scenarios beyond the zero-shot setup considered. Therefore, we believe that our work could contribute to broader ODQA scenarios in the future.

## C HUMAN EVALUATION OF GENERATED SUMMARIZATION

In this section, we provide details on the human preference evaluation of generated summarizations in Figure 4(c). First, we generate summarizations with a generic method (Listing 6) and with our proposed SURE (Listing 2). To separately consider the quality as rationale with the accuracy of prediction, we only compare the samples correctly predicted by both SURE and generic summarization; it results in 84 examples from the NQ*. Then, using the prompt in Listing 11, we conduct human evaluation. Specifically, we hired seven NLP experts off-line for our human evaluation experiment. Unlike asking GPT-4 with Listing 10, we ask human evaluators to answer as a tie if it is hard to determine.

Table 9: Few-shot experimental results. We use ChatGPT and $N = 10$ most relevant passages are commonly retrieved with BM25.

| EM / F1 | 0-shot | 1-shot | 5-shot |
|---|---|---|---|
| NQ*: Base | 29.4 / 41.7 | 30.1 / 39.3 | 31.9 / 42.0 |
| NQ*: SURE (Ours) | 35.6 / 44.9 | 36.3 / 46.8 | 37.2 / 47.7 |
| WebQ*: Base | 19.4 / 32.2 | 19.6 / 32.9 | 19.9 / 33.5 |
| WebQ*: SURE (Ours) | 23.2 / 36.5 | 24.2 / 39.4 | 24.3 / 38.5 |

**Listing 8** Prompt for answer candidates generation with few-shot examples.

```
f'''
Below are N passages related to the question at the end. We also provide
↪  the answers for various questions. After reading the passages and
↪  question-answer pairs, provide two correct candidates for the answer
↪  to the question at the end. Each answer should be in the form: (a)
↪  xx, (b) yy, and should not exceed 3 words for each candidate.

Passage #1 Title: {Passage #1 Title}
Passage #1 Text: {Passage #1 Text}

...

Passage #N Title: {Passage #N Title}
Passage #N Text: {Passage #N Text}

Question: {Example question #1}
Answer: {Example answer #1}

...

Question: {Example question #shot}
Answer: {Example answer #shot}

Question: {Query question}
Provide two correct candidates for the answer:
'''
```

# D    ADDITIONAL QUALITATIVE RESULTS

In this section, we present more qualitative results with SURE. All the examples are from NQ*, and ChatGPT with BM25 ($N = 10$) is commonly used.

## D.1    MORE EXAMPLES OF QUALITATIVE COMPARISON BETWEEN SURE'S SUMMARIZATION AND GENERIC SUMMARIZATION

In Figures 6, 7, and 8, we present more examples for qualitative comparison between the candidate-conditioned summarization by SURE and generic summarization. Innecessary and tedious sentences irrelevant to the answer are highlighted with **red**.

## D.2    QUALITATIVE EXAMPLES OF VERIFICATION WITH INSTANCE-WISE VALIDITY

To qualitatively show which samples are considered as *invalid* by LLMs, we present the examples that exhibit $v(s_k) = 0$ as $\mathcal{M}\left(p_{\text{val}}(q, y_k, s_k)\right) = \text{False}$ in Figure 9. Here, we highlight the sentences with **green** if they include the relevant context with the given candidate. In addition, we highlight the sentences with **red** if they induce a different candidate as an answer or do not support the candidate. For example, in the second example with a question (`Who is the actor that plays Saul on ''Grace and Frankie''?`), one can observe that the generated summarization

**Listing 9** Base prompt with few-shot examples.

```
f'''
Passage #1 Title: {Passage #1 Title}
Passage #1 Text: {Passage #1 Text}

...

Passage #N Title: {Passage #N Title}
Passage #N Text: {Passage #N Text}

Task description: predict the answer to the following question. Do not
↪  exceed 3 words.

Question: {Example question #1}
Answer: {Example answer #1}

...

Question: {Example question #K shot}
Answer: {Example answer #K shot}

Question: {Query question}
Answer:
'''
```

**Listing 10** Prompt for GPT-based evaluation.

```
f'''
Question: Given the following summaries for the target question,
↪  determine which one is more informative and plausible as rationale
↪  to support a given target question-answer pair.

Summary 1: {Summary 1}

Summary 2: {Summary 2}

Target Question: {Question}

Target Answer: {Answer}

Your Task:
Identify which summary (Summary 1 or Summary 2) is more informative and
↪  plausible as rationale to support a given answer at hand. Choices:
↪  [Summary 1, Summary 2].

Answer:
'''
```

concludes that the given candidate (`Mark Saul`) is incorrect; consequently, LLMs evaluates its validity as supporting summarization for the given candidate as false.

### D.3 Qualitative examples of verification with pair-wise ranking

In Figure 10, we present examples of verification by pair-wise ranking. Here, we highlight with **green** for the summarization that gets a higher ranking. In contrast, we highlight with **red** for the summarization that gets a lower ranking. We also highlight the relevant texts with the same colors, respectively.

---

**Listing 11** Template for human evaluation.

```
f'''
Given the following summaries for the target question, determine which
↪  one is more informative and plausible as rationale to support a
↪  given target question-answer pair.

Target Question: {Question}

Target Answer: {Answer}

Summary 1: {Summary 1}

Summary 2: {Summary 2}

Choices: [Summary 1, Tie, Summary 2]

Your choice:
'''
```

---

Table 10: Accuracy and cost ($) for each method. For the method in the last row, ChatGPT is used for Eq 2 and 3, and LLaMA is used for Eq 4 and 5, respectively.

| Exact Match (EM) | NQ* | WebQ* | Average Cost for API |
|---|---|---|---|
| Base (10 passages, ChatGPT) | 29.4 | 19.4 | 1.57$ |
| Base (50 passages, ChatGPT) | 33.8 | 21.8 | **7.67**$ |
| SURE (10 passages, ChatGPT) | **35.6** | 23.2 | 6.05$ |
| SURE (10 passages, ChatGPT + LLaMA) | 35.0 | **24.8** | 5.03$ |

## E DISCUSSION ON COST AND QUALITY GAIN

While SURE significantly improves QA system of LLMs, one can be concerned about its cost as it requires multiple inferences of LLMs. However, we note that the improvement of SURE is not just a simple consequence of more cost. Compared to other cost-increasing methods for accuracy improvement, SURE significantly outperforms them, *i.e.*, SURE is an even more efficient way to increase performance. For instance, increasing the number of retrieved passages is one of the most straightforward methods for this goal. But, in this case, SURE with 10 passages outperforms the base prompting with 50 passages, even with a lower total cost, as presented in Table 10. In addition, we note that other baseline approaches such as chain-of-thought or self-verification (considered in Table 1) also require more cost than base prompting, but they fail to successfully improve the performance.

On the other hand, one can reduce the overall cost by using cheaper LLMs for specific components, thanks to the modularity of SURE. Remarkably, SURE is compatible with the recent state-of-the-art open LLMs (see Tables 2 and 5) and hence this advantage is more noticeable. To give an intuition, we conduct the new experiments by using ChatGPT for the answer candidate generation and summarization, and LLaMA for the succeeding verification steps. As shown in the 4th row of Table 10, this hybrid approach of different LLMs with SURE successfully reduces the cost while keeping the effectiveness for improving the accuracy; for WebQ*, this approach even outperforms the expensive one. This result is from the effectiveness of LLaMA in WebQ* and indicates the potential of such a hybrid method.

Lastly, we further remark that most of SURE's cost is currently from re-reading retrieved passages for conditional summarizations (*e.g.*, 38% of the total cost for SURE with 10 passages). This is due to current APIs not providing recycling options for previous inputs. If recycling becomes available, SURE's cost could be significantly reduced.

## F    LIMITATION AND FUTURE WORK

In this work, we primarily focused on zero-shot setup for the experiments, which is a commonly encountered scenario in the real world, *e.g.*, search engine. But, similar to the previous works (Chowdhery et al., 2022; Touvron et al., 2023a), incorporating data-specific few-shot examples is also an interesting future direction to further improve QA accuracy of LLMs with SURE. Another interesting direction is extending the applied task beyond QA, such as language modeling (Guu et al., 2020) or language understanding tasks (Hendrycks et al., 2021).

## G    ADDITIONAL RELATED WORK

**Summarization in open-domain.** A summarization of retrieved passages has been considered in open-domain context; for example, there are recent works that propose to learn a module to selectively use the retrieved information in sentence- (Khattab et al., 2021; Su et al., 2022a) or passage-level (Mao et al., 2021; Chuang et al., 2023). In addition, Su et al. (2022a); Giorgi et al. (2023) form a new task that combines both passage retrieval and summarization for a given query, and Gao et al. (2023) considers summarization of information for prompting. However, these works require a large annotated dataset to extract the information specified to answer the question or construct the generic summarization which focuses on preserving the retrieved information within reduced texts.

## H    EXPERIMENTAL RESULTS WITH CONFIDENCE INTERVAL

In this section, we present confidence intervals for our main tables (Tables 1 and 2). To achieve this, we apply bootstrapping (Efron & Tibshirani, 1994), a popular technique for statistical inference that involves random sampling with replacement. We report 95% confidence intervals obtained through 1,000 iterations of bootstrapping. The confidence intervals for the EM and F1 metrics of each main table can be found in Tables 11, 12, 13, and 14.

The reliability of the results is reasonably robust, with the 95% confidence interval having only about a 10% variance from the reported value. Specifically, in the EM metric of the NQ dataset, our SuRe has the lowest confidence interval value at 32.0, compared to the maximum value of 29.1 for the no retrieval baseline and 30.0 for the best competitor. This demonstrates that the advantage of SuRe over prior works is statistically significant.

Table 11: EM with different QA methods with ChatGPT on four QA datasets. The 95% confidence intervals are calculated via bootstrapping by 1000 iterations, and presented below the corresponding values. $N = 10$ most relevant passages are retrieved using BM25, except *no retrieval*. The best and second best scores are highlighted in **bold** and underline, respectively.

| Methods / Datasets | NQ | WebQ | 2Wiki | HotpotQA |
|---|---|---|---|---|
| No retrieval | 27.6 | 25.0 | 21.4 | 22.2 |
| | [26.2, 29.1] | [23.2, 27.0] | [17.6, 25.2] | [18.8, 25.8] |
| Base | 28.4 | 19.6 | 27.4 | 30.8 |
| | [27.0, 30.0] | [17.9, 21.3] | [23.6, 31.0] | [26.6, 35.2] |
| Rerank | 24.8 | 18.8 | 23.0 | 27.8 |
| | [23.4, 26.2] | [17.2, 20.6] | [19.6, 26.8] | [23.8, 32.0] |
| RePlug | 26.0 | 18.8 | 23.6 | 28.0 |
| | [24.6, 27.4] | [17.1, 20.6] | [20.0, 27.2] | [24.2, 32.0] |
| Selection-inference | 24.3 | 17.3 | 22.6 | 30.8 |
| | [22.9, 25.7] | [15.7, 18.8] | [19.0, 26.0] | [26.8, 34.8] |
| Chain-of-thoughts | 22.3 | 15.2 | 19.6 | 25.6 |
| | [20.8, 23.6] | [13.7, 16.6] | [16.0, 23.2] | [21.6, 29.4] |
| Self-verification | 25.2 | 16.1 | 23.2 | 31.6 |
| | [23.7, 26.6] | [14.6, 17.7] | [19.6, 27.6] | [27.6, 35.8] |
| SuRe (Ours) | **33.5** | **25.1** | **32.8** | **33.2** |
| | [32.0, 35.0] | [23.1, 27.0] | [28.6, 36.8] | [29.0, 37.6] |

Table 12: F1 with different QA methods with ChatGPT on four QA datasets. The 95% confidence intervals are calculated via bootstrapping by 1000 iterations, and presented below the corresponding values. $N = 10$ most relevant passages are retrieved using BM25, except *no retrieval*. The best and second best scores are highlighted in **bold** and underline, respectively.

| Methods / Datasets | NQ | WebQ | 2Wiki | HotpotQA |
|---|---|---|---|---|
| No retrieval | 39.0 | **38.8** | 24.8 | 31.9 |
| | [37.7, 40.5] | [36.9, 40.7] | [21.1, 28.5] | [28.2, 35.4] |
| Base | 38.8 | 32.5 | 32.8 | 40.3 |
| | [37.3, 40.4] | [30.7, 34.3] | [28.9, 36.3] | [36.4, 44.4] |
| Rerank | 33.9 | 30.6 | 28.4 | 37.4 |
| | [32.5, 35.3] | [28.8, 32.4] | [25.1, 32.1] | [33.4, 41.4] |
| RePlug | 35.3 | 31.5 | 28.5 | 37.9 |
| | [33.9, 36.8] | [29.8, 33.3] | [24.7, 32.1] | [34.2, 41.9] |
| Selection-inference | 32.8 | 28.6 | 29.5 | 39.6 |
| | [31.5, 34.3] | [27.0, 30.3] | [26.2, 33.2] | [35.8, 43.7] |
| Chain-of-thoughts | 31.4 | 27.8 | 22.5 | 31.8 |
| | [30.1, 32.8] | [26.1, 29.4] | [18.8, 26.2] | [27.8, 35.7] |
| Self-verification | 35.4 | 28.5 | 30.5 | 41.8 |
| | [33.9, 36.9] | [26.7, 30.2] | [27.1, 34.9] | [37.8, 45.6] |
| SuRe (Ours) | **42.3** | 36.6 | **38.1** | **43.4** |
| | [40.8, 43.7] | [34.8, 38.5] | [34.0, 42.0] | [39.4, 47.6] |

Table 13: EM with different configurations of LLMs and retrieval methods on four QA datasets. The 95% confidence intervals are calculated via bootstrapping by 1000 iterations, and presented below the corresponding values. $N = 10$ most relevant passages are commonly retrieved. For LLaMA2-chat, we conducted experiments on NQ* and WebQ* and the results are indicated by *.

| Datasets | ChatGPT | | | | | | GPT-4 | | LLaMA2-chat | |
|---|---|---|---|---|---|---|---|---|---|---|
| | BM25 | + SURE | DPR | + SURE | Contriever | + SURE | BM25 | + SURE | BM25 | + SURE |
| NQ | 28.4 | **33.5** | 36.1 | **41.0** | 35.8 | **40.4** | 30.2 | **32.4** | 18.6* | **30.4*** |
| | [27.0, 30.0] | [32.0, 35.0] | [34.5, 37.7] | [39.4, 42.6] | [34.2, 37.4] | [38.8, 42.0] | [28.8, 31.7] | [30.9, 33.9] | [15.0, 22.0] | [26.4, 34.4] |
| WebQ | 19.6 | **25.1** | 23.2 | **27.3** | 22.5 | **28.7** | 21.5 | **21.7** | 16.0* | **24.0*** |
| | [17.8, 21.4] | [23.1, 27.0] | [21.4, 25.1] | [25.3, 29.2] | [20.5, 24.3] | [26.7, 30.7] | [19.6, 23.2] | [19.8, 23.4] | [13.0, 19.2] | [20.4, 27.8] |
| 2Wiki | 27.4 | **32.8** | 19.2 | **21.4** | 27.2 | **32.6** | 34.8 | **38.2** | 20.2 | **27.8** |
| | [23.6, 31.0] | [28.6, 36.8] | [15.8, 22.8] | [17.8, 24.8] | [23.2, 31.0] | [28.4, 36.8] | [30.6, 38.8] | [34.0, 42.2] | [16.6, 23.6] | [23.6, 32.0] |
| HotpotQA | 30.8 | **33.2** | 25.6 | **27.4** | 32.2 | **33.6** | 34.8 | **40.6** | 24.0 | **28.0** |
| | [26.6, 35.2] | [29.0, 37.6] | [21.8, 29.4] | [23.8, 31.4] | [28.2, 36.6] | [29.2, 37.6] | [31.0, 38.8] | [36.2, 44.6] | [20.2, 27.6] | [24.0, 32.0] |

Table 14: F1 with different configurations of LLMs and retrieval methods on four QA datasets. The 95% confidence intervals are calculated via bootstrapping by 1000 iterations, and presented below the corresponding values. $N = 10$ most relevant passages are commonly retrieved. For LLaMA2-chat, we conducted experiments on NQ* and WebQ* and the results are indicated by *.

| Datasets | ChatGPT | | | | | | GPT-4 | | LLaMA2-chat | |
|---|---|---|---|---|---|---|---|---|---|---|
| | BM25 | + SURE | DPR | + SURE | Contriever | + SURE | BM25 | + SURE | BM25 | + SURE |
| NQ | 38.8 | **42.3** | 47.4 | **50.8** | 47.6 | **50.4** | 40.9 | **42.4** | 36.4* | **42.9*** |
| | [37.3, 40.4] | [40.8, 43.7] | [45.9, 48.9] | [49.3, 52.4] | [46.2, 49.1] | [48.8, 51.8] | [39.5, 42.4] | [41.0, 43.9] | [32.8, 39.6] | [38.9, 46.8] |
| WebQ | 32.5 | **36.6** | 37.5 | **40.4** | 37.7 | **40.9** | **36.4** | 32.1 | 35.3* | **40.5*** |
| | [30.8, 34.3] | [34.8, 38.5] | [35.8, 39.3] | [38.5, 42.3] | [35.8, 39.6] | [39.1, 42.7] | [34.5, 38.2] | [30.2, 34.0] | [32.2, 38.7] | [36.7, 43.9] |
| 2Wiki | 32.8 | **38.1** | 22.8 | **25.2** | 32.9 | **37.1** | 39.2 | **43.2** | 31.2 | **36.2** |
| | [28.9, 36.3] | [34.0, 42.0] | [19.4, 26.3] | [21.6, 28.8] | [29.3, 36.9] | [33.0, 41.4] | [35.2, 43.0] | [39.3, 47.2] | [27.7, 34.9] | [32.2, 39.9] |
| HotpotQA | 40.3 | **43.4** | 34.6 | **35.5** | 42.9 | **43.5** | 44.4 | **50.4** | **39.6** | 38.4 |
| | [36.4, 44.4] | [39.4, 47.6] | [30.8, 38.1] | [31.6, 39.6] | [36.6, 47.2] | [39.3, 47.7] | [40.5, 48.1] | [46.3, 54.3] | [35.9, 43.2] | [34.6, 42.5] |

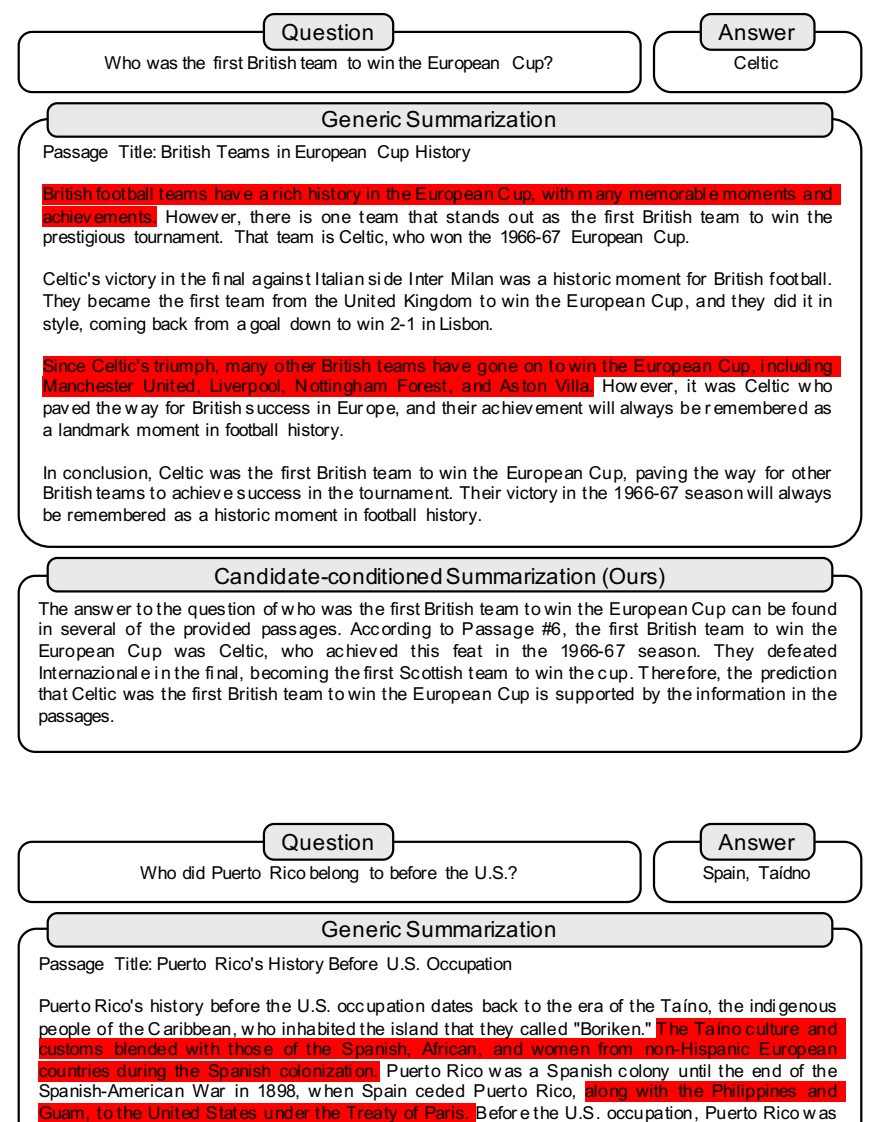

Figure 6: Qualitative comparison of candidate-conditioned summarization from SURE (Ours) compared to generic summarization as a rationale for the answer.

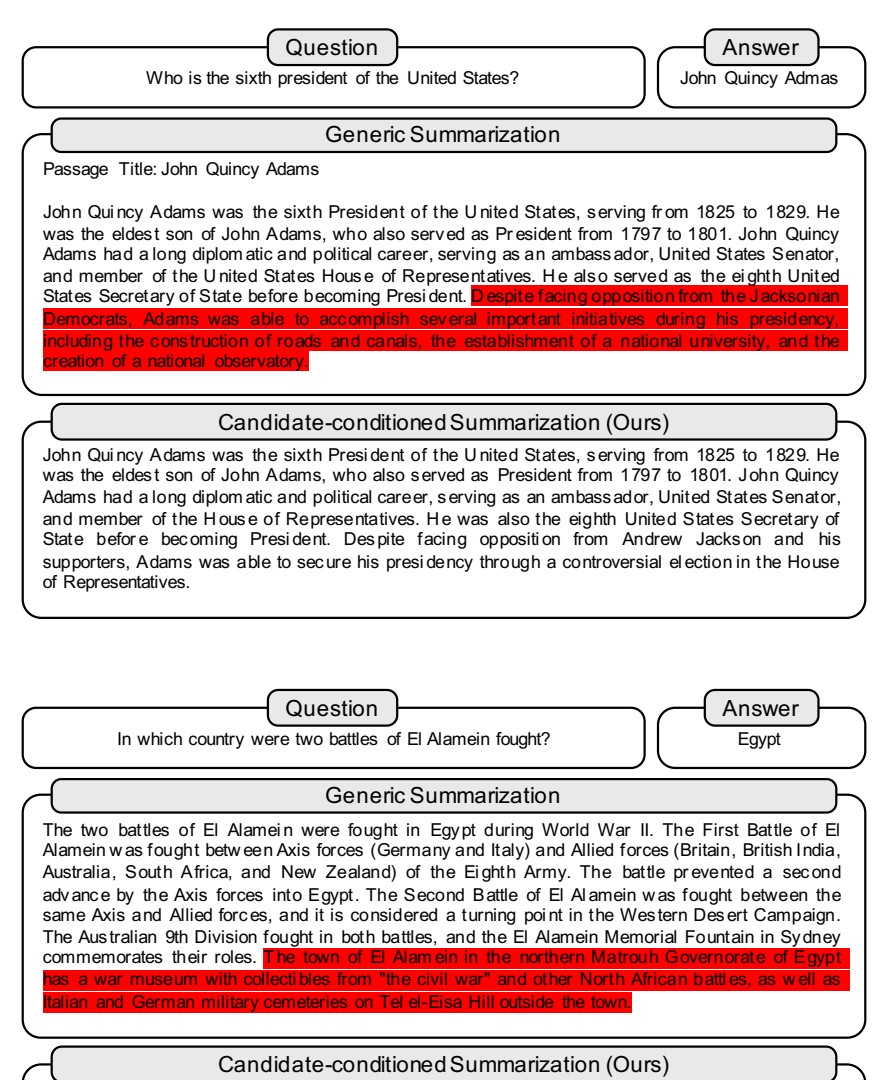

Figure 7: Qualitative comparison of candidate-conditioned summarization from SuRe (Ours) compared to generic summarization as a rationale for the answer.

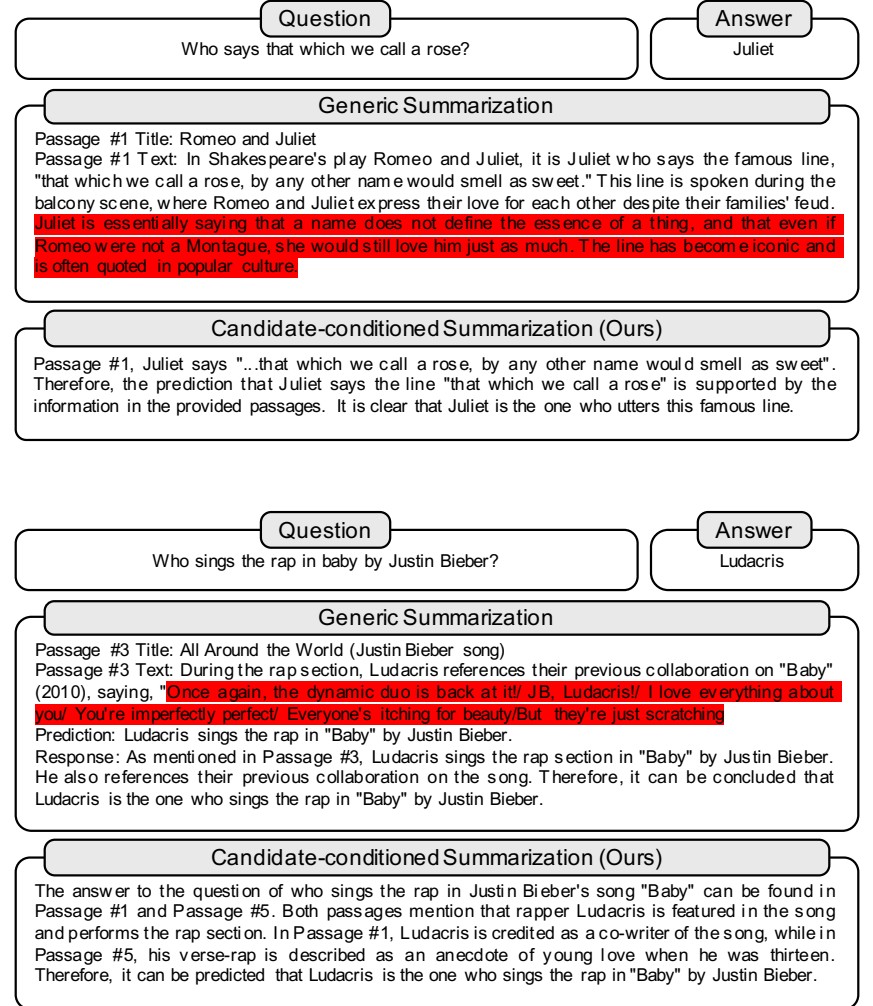

Figure 8: Qualitative comparison of candidate-conditioned summarization from SURE (Ours) compared to generic summarization as a rationale for the answer.

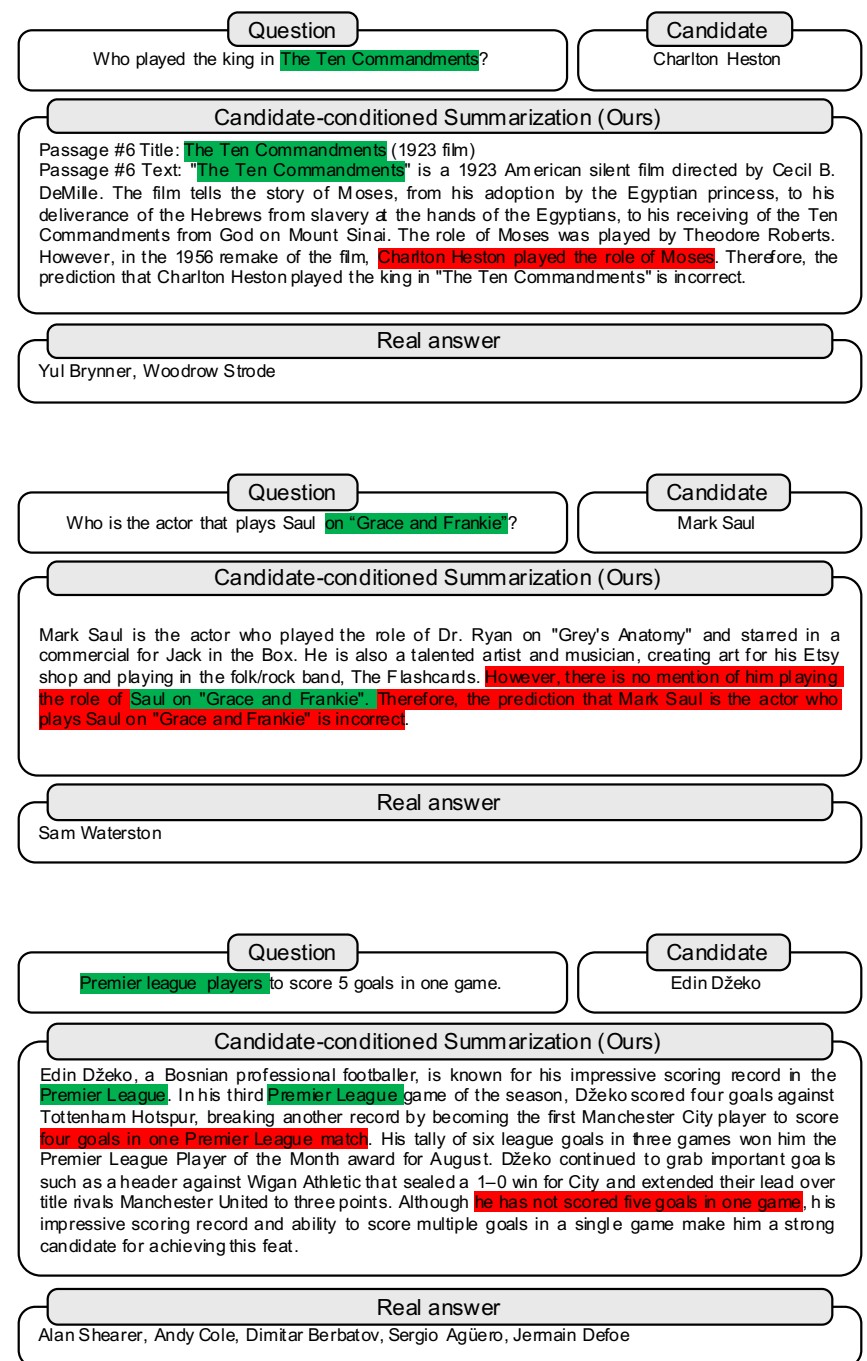

Figure 9: Example summarizations that are evaluated as invalid by LLMs (Eq. 4).

**Question**

Who is the CEO of Crate and Barrel?

**Candidate-conditioned Summarization 1 (Candidate: Neela Montgomery)**

Neela Montgomery is the current CEO of Crate & Barrel, having assumed the role after Doug Diemoz departed in April 2017. Montgomery brings a wealth of experience to the position, having previously served as the Executive Board Member for Multi-Channel Retail at Otto Group, the parent company of Crate & Barrel. Under her leadership, the company is expected to continue its expansion into new markets, both domestically and internationally. Montgomery's focus on innovation and customer experience is sure to drive growth and success for Crate & Barrel in the years to come.

**Candidate-conditioned Summarization 2 (Candidate: Sascha Bopp)**

Sascha Bopp was named as the CEO of Crate & Barrel after Barbara Turf retired in 2012. However, due to poor performance during the 2013 holiday season and growing disenchantment from employees, Bopp was removed from his position and terminated from the company. Since then, there have been several other CEOs, including Doug Diemoz and currently Neela Montgomery. However, based on the information provided, it can be predicted that Sascha Bopp was the CEO of Crate & Barrel at some point in time.

**Real answer**

Neela Montgomery

**Question**

The concept of micro financing was developed by who in 1976?

**Candidate-conditioned Summarization 1 (Candidate: Muhammad Yunus)**

Microfinance is a term used to describe financial services, such as loans, savings, and insurance, provided to low-income individuals or those who do not have access to traditional banking services. The concept of microfinance has been around for decades, but it was not until the 1970s that it gained widespread attention. In 1976, Muhammad Yunus, a Bangladeshi economist, began experimenting with lending small amounts of money to poor women in rural areas. He founded the Grameen Bank in 1983, which became the first microfinance institution to provide small loans to the poor without requiring collateral. Yunus and the Grameen Bank were awarded the Nobel Peace Prize in 2006 for their efforts to create economic and social development through microfinance. Therefore, it can be concluded that the concept of micro financing was developed by Muhammad Yunus in 1976.

**Candidate-conditioned Summarization 2 (Candidate: Grameen Bank)**

Microfinance is a term that has evolved over time, from micro-credit to micro-finance. It is a way to promote economic development, employment, and growth through the support of micro-entrepreneurs and small businesses. The Grameen Bank is one of the pioneers of microfinance, having been founded in Bangladesh in 1983 by Muhammad Yunus. The bank's mission was to provide small loans to poor people who did not have access to traditional banking services. The bank's success in reducing poverty through microfinance has inspired similar initiatives around the world. While microfinance has its critics, it has undoubtedly helped many people lift themselves out of poverty. Therefore, it can be concluded that the concept of micro financing was not developed by Muhammad Yunus in 1976, but rather by the Grameen Bank in 1983.

**Real answer**

Muhammad Yunus

Figure 10: Example summarizations with pair-wise rank evaluation (Eq. 5).

