# OpenReview forum: "SuRe: Summarizing Retrievals using Answer Candidates for Open-domain QA of LLMs"
_ICLR.cc/2024/Conference — ICLR 2024 poster_

### Official Review · Reviewer_MmoS · 2023-10-28

**Soundness:** 2 fair
**Presentation:** 3 good
**Contribution:** 2 fair
**Rating:** 6
**Confidence:** 4

**Summary:**

The paper presented a new design, named SuRE, of combining retrieval and prompting to improve LLM's Open Book QA quality. It starts with retrieving relevant documents about the question, then generates a list of candidate answers with vanilla retrieval-augmented generation (RAG) prompting. The key contributions then follow. It prompts LLMs to produce a per-candidate summary of the retrieved documents to support the candidate. Then, LLM-based point-wise and pair-wise critiques are used to assess the quality of the per-candidate supporting summary. The candidate with the most sound supporting summary is chosen as the final answer. This is based on an intuition that the supporting summary for a correct candidate is usually of higher quality.

The paper designed experiments to show the quality gain of SuRe, which surpasses vanilla RAG and other recent algorithms of improvement. It also showed that the improvement is consistent with different retrieval algorithms and LLMs. Ablation study is run to understand the contribution of different part of SuRE. The paper thus concludes that SuRe is an effective way to improve retrieval-based open book QA without the need to finetune the underlying LLM.

**Strengths:**

- Writing and clarity. The paper is mostly well-written, easy to follow, and free of grammar / formatting errors.
- Comprehensive experiment design. The paper tested SuRe with different datasets, retrieval algorithms and LLMs as those factors could have a big impact on the outcome.
- Reviewer particularly likes the way the paper listing all research questions explicitly at the beginning of the experiment section with reference to corresponding tables and figures: clear and easy to follow.

**Weaknesses:**

Here the Reviewer tries to order the weakness by their priority.

- The title and initial claim in the abstract are too broad. They almost sound like a claim of re-inventing RAG. Abstract called out of hallucination and grounding, but it's not specifically studied in the paper, at least not more than just exact-match rate and F1 scores. It would be better if the authors make the title and abstract more specific to the contribution.

- Efficiency and cost, which is an important shortcoming of SuRe, is not discussed. During the process, SuRe makes a significant amount of calls to the underlying LLM. They are both slow and costly (in direct money terms in the case of calling commercial APIs). It would be useful to show the comparison and let any potential users know the cost of the quality gain.

- Ablation study is poorly designed. Reviewer is expecting a study where each of the key component of SuRe  is removed (ablated). However the paper showed the results of each one individually added. The subcomponents are not 1 to 1 mapped to SuRe either. For example what is MCQ and how it maps back to SuRe? Some readers may figure it out eventually but it's hard for the Reviewer to get it in a short time.

- Need to be specific about "limitedly explored" when talking about previous work. A potential reader should not need to guess or read the whole reference paper to understand where the limits of the previous exploration are in the author's view.

See more trivial comments related to weakness in the Questions section.

**Questions:**

- Section 1 paragraph 2: "implicitly forced to use the retrieved information without additional training". The Reviewer didn't get it. Do you mean use or not use?

- Numbers in the experiment section: It's better to give some confidence interval of the EM/F1 numbers since they are obtained on a smaller sample (500, if the Reviewer recalls correctly).

- Section 4.3: Expand "MCQ" and explain what Robinson et al (2023) did. Readers should not have to read a reference paper if they don't want to dive deeper.

- Figure 4 (a): The shape of the point is hard to read, and only using the red-blue color to distinguish lines could be a problem for color-blind people. (Reviewer appreciates the effort of adding patterns to (b) and (c) so color is not the only discriminator).

- Last paragraph of Section 4. Reviewer suggest adding the number of human-preference samples (84) here, saving readers a trip to the Appendix.

**Details Of Ethics Concerns:**

None.

---

> ### Author Response · Authors · 2023-11-20
> **Response to Reviewer MmoS (1/3)**
>
> Dear Reviewer MmoS,
>
> We sincerely appreciate your review with thoughtful comments. We have carefully considered each of your questions and provide detailed responses below. Please let us know if you have any further questions or concerns.
>
> ---
>
> **[W1] The title and initial claim in the abstract are too broad**
>
> Thank you for making constructive suggestions to improve the clarity of our manuscript. To clarify our contribution, we revised the title and abstract to be more specific. Specifically, we specified our core component in the title. In addition, we removed a calling out of hallucination and grounding, and more focused on our contribution to QA task in the abstract. For your convenience, we present the revised title and abstract below. Here, we highlight the revised sentences in **bold** and the original ones in *italics* within parentheses:
> - (**Title**) **SuRe: Summarizing Retrievals using Answer Candidates for Open-domain QA of LLMs** (before: *SuRe: Improving Open-domain Question Answering of LLMs via Summarized Retrieval*)
> - (**Abstract**) **Large language models (LLMs) have made significant advancements in various natural language processing tasks, including question answering (QA) tasks.** (before: *Large language models (LLMs) have made significant advancements in various natural language processing tasks, but face challenges such as hallucinations and integration of up-to-date knowledge, which is particularly critical for question answering (QA).*) While incorporating new information with the retrieval of relevant passages is a promising way to improve QA with LLMs, the existing methods often require additional fine-tuning which becomes infeasible with recent LLMs. **Augmenting retrieved passages of QA via prompting** (before: *Retrieval augmentation via prompting*) has the potential to address this limitation, but this direction has been limitedly explored. To this end, we design a simple yet effective framework to enhance open-domain QA (ODQA) with LLMs, based on the summarized retrieval (SuRe). SuRe helps LLMs predict **more accurate answers for a given question** (before: *more grounded answers*), which are well-supported by the summarized retrieval that could be viewed as an explicit rationale extracted from the retrieved passages. Specifically … (continued)
>
> ---
>
> **[W2] Efficiency and cost … It would be useful to show the comparison and let any potential users know the cost of the quality gain**
>
> First, we clarify that the improvement of SuRe is not just a simple consequence of more cost. Compared to other cost-increasing methods for accuracy improvement, SuRe significantly outperforms them, i.e., SuRe is an even more efficient way to increase performance. For instance, increasing the number of retrieved passages is one of the most straightforward methods for this goal. But, in this case, SuRe with 10 passages outperforms the base prompting with 50 passages, even with a lower total cost, as presented in the below table (see 1st~3rd rows, values are from Figure 4). In addition, we note that other baseline approaches such as chain-of-thought or self-verification (considered in Table 1) also require more cost than base prompting, but they fail to successfully improve the performance.
>
>
> \begin{array}{c|cc|c}
> \hline
> \text{Exact Match (EM)} & \text{NQ*} & \text{WebQ*} & \text{Average Cost for API} \newline \hline
> \text{Base (10 passages, ChatGPT)} & 29.4 & 19.4 & 1.57\$ \newline
> \text{Base (50 passages, ChatGPT)} & 33.8 & 21.8 & \textbf{7.67}\$ \newline
> \text{SuRe (10 passages, ChatGPT)} & \textbf{35.6} & 23.2 & 6.05\$ \newline \hline
> \text{SuRe (10 passages, ChatGPT (Eq. 2 and Eq.3) + LLaMA (Eq.4 and Eq.5))} & 35.0 & \textbf{24.8} & 5.03\$ \newline \hline
> \end{array}
>
> On the other hand, one can reduce the overall cost by using cheaper LLMs for specific components, thanks to the modularity of SuRe. Remarkably, SuRe is compatible with the recent state-of-the-art open LLMs (see Tables 2 and 5) and hence this advantage is more noticeable. To give an intuition, we conduct the new experiments by using ChatGPT for the answer candidate generation and summarization, and LLaMA for the succeeding verification steps. As shown in the 4th row of the above table, this hybrid approach of different LLMs with SuRe successfully reduces the cost while keeping the effectiveness for improving the accuracy; for WebQ*, this approach even outperforms the expensive one. This result is from the effectiveness of LLaMA in WebQ* and indicates the potential of such a hybrid method.
>
> Lastly, we further remark that most of SuRe's cost is currently from re-reading retrieved passages for conditional summarizations (e.g., 38% of the total cost for SuRe with 10 passages). This is due to current APIs not providing recycling options for previous inputs. If recycling becomes available, SuRe's cost could be significantly reduced. We added this discussion in Appendix E of the revised draft.

---

> ### Author Response · Authors · 2023-11-20
> **Response to Reviewer MmoS (2/3)**
>
> **[W3] Ablation study is poorly designed … The subcomponents are not 1 to 1 mapped to SuRe either…**
>
> We first denote that our original ablation study included necessary experiments to analyze the effectiveness of SuRe. However, we recognize that our initial presentation may not have clearly conveyed our intentions. To clarify, we have revised our table for ablation (Table 3) to clearly associate each subcomponent with SuRe, as presented below. Specifically, we have renamed and reordered the ablated methods for easier understanding and added an additional method, "*+ Conditional summarizations*" accompanied by conducting new experiments. We have also distinguished the methods used for further analysis (*MCQ prompt* & *Sum-and-pred (Gen)*) from the methods for the ablation study. In precise, these two methods are for a detailed investigation into SuRe's improvement source, not for ablation study. We have stated this explicitly and updated the corresponding descriptions in Section 4.3 of the revised draft.
>
> \begin{array}{l|cccc|c}
> \hline
> \text{Methods / Datasets} & \text{NQ*} & \text{WebQ*} & \text{2Wiki} & \text{HotpotQA} & \text{Average} \newline \hline
> \text{Base} & 29.4 / 41.7 & 19.4 / 32.2 & 28.4 / 33.2 & 30.8 / 40.3 & 27.0 / 36.9 \newline
> \text{+Conditional summarizations} & 30.4 / 40.9 & 20.8 / 33.5 & 29.2 / 34.5 & 33.0 / \textbf{43.4} & 28.4 / 38.1 \newline
> \text{+Pair-wise ranking} & 30.6 / 41.2 & 21.6 / 34.8 & 31.0 / 36.0 & 30.6 / 40.7 & 28.5 / 38.2  \newline
> \text{+Instance-wise validity (SuRe)}  & \textbf{35.6} / {44.9} & \textbf{23.2} / \textbf{36.5} & \textbf{32.8} / \textbf{38.1} & {33.2} / \textbf{{43.4}} & \textbf{31.2} / \textbf{40.7} \newline \hline
> \text{MCQ prompt} & {35.2} / \textbf{45.3} & 22.4 / 35.1 & 30.4 / 36.1 & 31.0 / 41.5  & 29.8 / 39.5 \newline
> \text{Sum-and-pred (Gen)} & 26.4 / 37.8 & 19.8 / 32.6 & 25.6 / 32.3 & \textbf{{33.8}} / {43.3}& 27.3 / 37.1 \newline \hline
> \end{array}
>
> ---
>
> **[W4] Need to be specific about "limitedly explored" when talking about previous work**
>
> Thank you for your insightful feedback. We understand the need for clarity when referring to 'limitedly explored' aspects of our work. In our context, 'limitedly explored' refers to the specific limitations in using retrieved information for prompting LLMs. Common practices like appending retrieved passages for prompting [1,2] have shown limitations in fully leveraging the information within these passages [3]. Another explored method is aggregating predictions from each retrieved passage [4,5]; however, this approach also has a notable limitation as LLMs struggle to comprehend the full context of the retrieved information.
>
> To further alleviate your concerns, we have revised the related work section to specify where the limits of the previous exploration are and included more prior works, as follows:
> - (**Original**) However, this direction has been only limitedly explored (Shi et al., 2023); so this work tries to investigate such research direction.
> - (**Revised**) However, this direction has been only limitedly explored. Appending the retrieval (Si et al., 2023; Trivedi et al., 2023) is a common practice for prompting, but Liu et al. (2023b) recently revealed its limitation in utilizing the retrieved information. Aggregating the predictions from each retrieved passage has been also explored (Lazaridou et al., 2022; Shi et al., 2023), but LLMs can’t see a full context of retrieved information in this case. On the other hand, the summarization of retrieved passages has been considered in open-domain context; for example, there are recent works that propose to learn a module to selectively use the retrieved information in sentence- (Khattab et al., 2021; Su et al., 2022) or passage-level (Mao et al., 2021; Chuang et al., 2023). In addition, Su et al. (2022); Giorgi et al. (2023) formed a new task that combines both passage retrieval and summarization for a given query, and Gao et al. (2023) consider summarization of information for prompting. However, these works require a large annotated dataset to extract the information specified to answer the question or construct the generic summarization which focuses on preserving the retrieved information within reduced texts.

---

> ### Author Response · Authors · 2023-11-20
> **Response to Reviewer MmoS (3/3)**
>
> **[Q1] Section 1 paragraph 2: "implicitly forced to use the retrieved information without additional training". The Reviewer didn't get it. Do you mean use or not use?**
>
> We intended to describe that LLMs use the retrieved information via prompting but has a certain limitation, since LLMs are simply instructed to use the retrieved information instead of being explicitly trained to use it. We agree that the current draft needs to be more clarified, and we modified the corresponding part in the revision (modified parts are highlighted in bold and original parts in italic within parentheses):
> - However, naive prompting could be limited in fully exploiting the retrieved contexts, **since LLMs are simply instructed to use the retrieved information instead of being explicitly trained to use it; for example,** (before: *as LLMs are implicitly forced to use the retrieved information without additional training*) Liu et al. (2023b) recently observed that LLMs struggle to handle long input contexts when they are naively appended.
>
> ---
>
> **[Q2] Numbers in the experiment section: It's better to give some confidence interval of the EM/F1 numbers since they are obtained on a smaller sample**
>
> We first clarify that we used a temperature of 0.0 when calling the API or greedy decoding when using LLaMA, as denoted in the implementation details in Section 4.1. Consequently, our experimental results are deterministic, and hence we do not present the confidence interval. In addition, we would like to remark that other works using a temperature of 0.0 report the experimental results in a similar way [6,7,8]. We further emphasize this in Section 4.1 of the revised draft.
>
> ---
>
> **[Q3] Section 4.3: Expand "MCQ" and explain what Robinson et al (2023) did. Readers should not have to read a reference paper if they don't want to dive deeper.**
>
> MCQ is an abbreviation of Multiple Choice Question, and Robinson et al (2023) [9] have demonstrated that multiple-choice prompts generally elicit much more accurate responses than do cloze prompts, for LLMs with high multiple-choice symbol binding ability like OpenAI Codex. Motivated by this, we consider MCQ prompt to further evaluate the effectiveness of our framework whether selecting the answer from the construction and verification of the conditional summarizations is more effective than direct prompting, under the same generated answer candidates (from Eq.2). In response to your comment, we revised the draft to expand MCQ and add more descriptions about this in Section 4.3, as follows:
> - (**Original**) MCQ prompt: generates the answer candidate via prompting (Eq. 2), and utilizes it as constraint for prediction by appending it as additional input similar to Robinson et al. (2023)
> - (**Revised**) MCQ prompt: composes Multiple Choice Questions by generating the answer candidates via prompting (Eq. 2) and using them as possible choices for prediction by appending them to input prompt (Robinson et al., 2023) (more details in Appendix A.7)
>
> In addition, we provided more explanations of MCQ prompt including what Robinson et al (2023) did and the used prompt in Appendix A.7.
>
> ---
>
> **[Q4] Figure 4 (a): The shape of the point is hard to read, and only using the red-blue color to distinguish lines could be a problem for color-blind people. (Reviewer appreciates the effort of adding patterns to (b) and (c) so color is not the only discriminator).**
>
> Thank you for the comments to improve the visualization of our manuscript for various readers. To address your concerns, we re-drew Figure 4 (a) by differentiating the shapes of the points across different datasets and methods, and enabled them to be distinguished regardless of color. To indicate this modification, we colorized the corresponding title of Figure 4 (a) in the revised draft.
>
> ---
>
> **[Q5] Last paragraph of Section 4. Reviewer suggest adding the number of human-preference samples (84) here, saving readers a trip to the Appendix.**
>
> Following your suggestion, we revised the draft to explicitly denote the number of human-preference samples (84) in the last paragraph of Section 4 and the corresponding caption of Figure 4, as follows (added sentences are highlighted in bold):
> - (**Last paragraph of Section 4**) we only compare the samples correctly predicted by both SuRe and Generic summarization used in Table 3; **for example, it results in 84 remaining samples in the case of NQ***. … Next, we ask human evaluators which summarization is more informative and plausible to support the given question-answer pair on **84 samples of NQ***.
> - (**Figure 4**) (c) Human preference between SuRe’s summarization and generic summarization on **84 samples of** NQ*.
>
> ---
>
> If you have any further questions/concerns, please do not hesitate to let us know.
>
> Thank you very much,
> Authors

---

> > ### Comment · Reviewer_MmoS · 2023-11-22
> > **Thank you for the detailed answers.**
> >
> > Most of the questions have been answered.
> >
> > Reviewer still have the question on the confidence interval: it's not about the undeterministic result, but about the meaningfulness of small changes in the evaluation.

---

> > > ### Author Response · Authors · 2023-11-23
> > > **Response to Reviewer MmoS**
> > >
> > > Dear Reviewer MmoS,
> > >
> > > We are glad to hear that we have addressed most of your questions. Also, thank you for raising the score!
> > >
> > > Regarding your remaining question about confidence intervals, we calculated 95% confidence intervals for our main tables (Tables 1 and 2) using bootstrapping [1] with 1,000 iterations, and presented the results in Tables 11 to 14 in Appendix G. The reliability of the results is reasonably robust, with the 95% confidence interval having only about a 10% variance from the reported value. Specifically, in the EM metric of the NQ dataset, our SuRe has the lowest confidence interval value at 32.0, compared to the maximum value of 29.1 for the no retrieval baseline and 30.0 for the best competitor. This demonstrates that the advantage of SuRe over prior works is statistically significant.
> > >
> > > Please let us know if you have any further questions.
> > >
> > > ---
> > >
> > > [1] Efron & Tibshirani. An Introduction to the Bootstrap. 1994.

---

> ### Author Response · Authors · 2023-11-20
> **References**
>
> [1] Si et al., Prompting GPT-3 to be Reliable., ICLR 2023
> [2] Trivedi et al., Interleaving Retrieval with Chain-of-thought Reasoning for Knowledge-intensive Multi-step Question., ACL 2023
> [3] Liu et al., Lost in the middle: How Language Models Use Long Contexts., arXiv:23.07
> [4] Lazaridou et al., Internet-augmented Language Models through Few-shot prompting for Open-domain Question Answering., arXiv:22.03
> [5] Shi et al., Replug: Retrieval-augmented Black-box Language Models., EMNLP 2023 Findings
> [6] Sun et al., Is ChatGPT Good at Search? Investigating Large Language Models as Re-Ranking Agents., EMNLP 2023
> [7] Yue et al., Automatic Evaluation of Attribution by Large Language Models., EMNLP 2023 Findings
> [8] Liang et al., Code as Policies: Language Model Programs for Embodied Control., ICRA 2023
> [9] Robinson et al., Leveraging Large Language Models for Multiple Choice Question Answering., ICLR 2023

---

### Official Review · Reviewer_AgBt · 2023-10-31

**Soundness:** 4 excellent
**Presentation:** 4 excellent
**Contribution:** 3 good
**Rating:** 8
**Confidence:** 4

**Summary:**

This paper proposes Summarized Retrieval (SURE) for open-domain QA. First, it generates answer candidates from retrieved passages  with LLMs. Then, for each candidate answer, it conditionally summarizes the retrieved passages in order to focus on extracting the candidate-relevant contexts. Then those answers are ranked by a weighted score of instance-wise validity score and pairwise informativeness score. Experimental results show that SURE significantly outperforms the baselines on multiple datasets and LLM configurations. Detailed ablation studies are also performed.

**Strengths:**

- A novel framework is proposed to enhance open-domain QA with LLMs where the candidate answer can be better grounded on retrieved passages.
- The experiments are well-conducted and the performance improvement is significant and consistent.
- The paper is very well-written.

**Weaknesses:**

- The proposed method could be expensive considering eq 4-6.

**Questions:**

The paper is clearly written. No more questions from me.

---

> ### Author Response · Authors · 2023-11-20
> **Response to Reviewer AgBt**
>
> Dear Reviewer AgBt,
>
> We sincerely appreciate your review with thoughtful comments. We have carefully considered each of your questions and provide detailed responses below. Please let us know if you have any further questions or concerns.
>
> ---
>
> **[W1] The proposed method could be expensive considering eq 4-6**
>
> We first clarify that Eq. 4-6 are relatively cheap as these processes do not include the retrieved passages as inputs; for example, these components consume 13% of the total cost of SuRe.
> Next, we highlight that the improvement of SuRe is not just a simple consequence of more cost. Compared to other cost-increasing methods for accuracy improvement, SuRe significantly outperforms them, i.e., SuRe is an even more efficient way to increase performance. For instance, increasing the number of retrieved passages is one of the most straightforward methods for this goal. But, in this case, SuRe with 10 passages outperforms the base prompting with 50 passages, even with a lower total cost, as presented in the below table (see 1st~3rd rows, values are from Figure 4). In addition, we note that other baseline approaches such as chain-of-thought or self-verification (considered in Table 1) also require more cost than base prompting, but they fail to successfully improve the performance.
>
>
>
> \begin{array}{c|cc|c}
> \hline
> \text{Exact Match (EM)} & \text{NQ*} & \text{WebQ*} & \text{Average Cost for API} \newline \hline
> \text{Base (10 passages, ChatGPT)} & 29.4 & 19.4 & 1.57\$ \newline
> \text{Base (50 passages, ChatGPT)} & 33.8 & 21.8 & \textbf{7.67}\$ \newline
> \text{SuRe (10 passages, ChatGPT)} & \textbf{35.6} & 23.2 & 6.05\$ \newline \hline
> \text{SuRe (10 passages, ChatGPT (Eq. 2 and Eq.3) + LLaMA (Eq.4 and Eq.5))} & 35.0 & \textbf{24.8} & 5.03\$ \newline \hline
> \end{array}
> On the other hand, one can reduce the overall cost by using cheaper LLMs for specific components, thanks to the modularity of SuRe. Remarkably, SuRe is compatible with the recent state-of-the-art open LLMs (see Tables 2 and 5) and hence this advantage is more noticeable. To give an intuition, we conduct the new experiments by using ChatGPT for the answer candidate generation and summarization, and LLaMA for the succeeding verification steps. As shown in the 4th row of the above table, this hybrid approach of different LLMs with SuRe successfully reduces the cost while keeping the effectiveness for improving the accuracy; for WebQ*, this approach even outperforms the expensive one. This result is from the effectiveness of LLaMA in WebQ* and indicates the potential of such a hybrid method.
>
> Lastly, we further remark that most of SuRe's cost is currently from re-reading retrieved passages for conditional summarizations (e.g., 38% of the total cost for SuRe with 10 passages). This is due to current APIs not providing recycling options for previous inputs. If recycling becomes available, SuRe's cost could be significantly reduced. We added this discussion in Appendix E of the revised draft.
>
>
> ---
>
> If you have any further questions/concerns, please do not hesitate to let us know.
>
> Thank you very much,
> Authors

---

> > ### Comment · Reviewer_AgBt · 2023-11-22
> >
> > Thanks for the authors' response ! My question is answered.

---

> > > ### Author Response · Authors · 2023-11-23
> > > **Response to Reviewer AgBt**
> > >
> > > Dear Reviewer AgBt,
> > >
> > > We are happy to hear that our rebuttal successfully answered your question! Please don't hesitate if you have any further questions.

---

### Official Review · Reviewer_9TBy · 2023-11-05

**Soundness:** 3 good
**Presentation:** 3 good
**Contribution:** 3 good
**Rating:** 6
**Confidence:** 4

**Summary:**

This work introduces Summarized Retrieval (SURE) to enhance the performance of Open-Domain Question Answering (ODQA) using retrieval-augmented Language Models (LLMs). The goal is to provide more well-grounded answers with LLMs by generating summarizations of retrieved passages, which serve as explicit rationales for the answers. By constructing multiple summarizations for each possible answer candidate, LLMs can then focus on context relevant to the candidate and provide more discriminative viewpoints for the question. Experiments are conducted on multiple QA datasets showing that SURE improves across all of them.

**Strengths:**

The idea of constructing the summaries of the retrieved passages for the potential answer candidates is somehow simple yet effective. The paper shows significant improvements across various datasets.

**Weaknesses:**

While SURE shows interesting results, there are some points that in my opinion should be clarified/improved before publication:
- It is unclear how this approach can scale. SURE may work well in experiments, but in real-world applications, the number of relevant passages can vary greatly and the various steps in SURE can become extremely costly, limiting the usefulness of this approach.
- The evaluation metrics are based on term overlaps and they might not capture all dimensions of model performance. Other factors like response coherence, relevance, and efficiency should also be considered, especially in the case of LLMs.

**Questions:**

- Why limit the evaluation to only EM/F1 and not consider LLMs approaches for automatic evaluation?
- Why only short-answer datasets? Have you considered long-answers? What would change in that case?

---

> ### Author Response · Authors · 2023-11-20
> **Response to Reviewer 9TBy (1/2)**
>
> Dear Reviewer 9TBy,
>
> We sincerely appreciate your review with thoughtful comments. We have carefully considered each of your questions and provide detailed responses below. Please let us know if you have any further questions or concerns.
>
> ---
>
> **[W1] It is unclear how this approach can scale … the number of relevant passages can vary greatly and the various steps in SURE can become extremely costly.**
>
> First, we note that our framework is compatible with the popular approaches to resolve the scalability issue of ODQA system, regarding a large number of relevant passages such as re-ranking [1,2] or multi-step retrieve-and-read [3,4]. Also, we remark that our framework consists of multiple components, and each component only gets the same number of passages compared to the standard prompting method. Thus, SuRe does not pose a specific scalability issue related to the number of relevant passages.
>
> In addition, compared to other cost-increasing methods for accuracy improvement, SuRe significantly outperforms them, i.e., SuRe is an even more efficient way to increase performance. For instance, increasing the number of retrieved passages is one of the most straightforward methods for this goal. But, in this case, SuRe with 10 passages outperforms the base prompting with 50 passages, even with a lower total cost, as presented in the below table (see 1st~3rd rows, values are from Figure 4). In addition, we note that other baseline approaches such as chain-of-thought or self-verification (considered in Table 1) also require more cost than base prompting, but they fail to successfully improve the performance.
>
> \begin{array}{c|cc|c}
> \hline
> \text{Exact Match (EM)} & \text{NQ*} & \text{WebQ*} & \text{Average Cost for API} \newline \hline
> \text{Base (10 passages, ChatGPT)} & 29.4 & 19.4 & 1.57\$ \newline
> \text{Base (50 passages, ChatGPT)} & 33.8 & 21.8 & \textbf{7.67}\$ \newline
> \text{SuRe (10 passages, ChatGPT)} & \textbf{35.6} & 23.2 & 6.05\$ \newline \hline
> \text{SuRe (10 passages, ChatGPT (Eq. 2 and Eq. 3) + LLaMA (Eq. 4 and Eq. 5))} & 35.0 & \textbf{24.8} & 5.03\$ \newline \hline
> \end{array}
>
> On the other hand, one can reduce the overall cost by using cheaper LLMs for specific components, thanks to the modularity of SuRe. Remarkably, SuRe is compatible with the recent state-of-the-art open LLMs (see Tables 2 and 5) and hence this advantage is more noticeable. To give an intuition, we conduct the new experiments by using ChatGPT for the answer candidate generation and summarization, and LLaMA for the succeeding verification steps. As shown in the 4th row of the above table, this hybrid approach of different LLMs with SuRe successfully reduces the cost while keeping the effectiveness for improving the accuracy; for WebQ*, this approach even outperforms the expensive one. This result is from the effectiveness of LLaMA in WebQ* and indicates the potential of such a hybrid method.
>
> Lastly, we further remark that most of SuRe's cost is currently from re-reading retrieved passages for conditional summarizations (e.g., 38% of the total cost for SuRe with 10 passages). This is due to current APIs not providing recycling options for previous inputs. If recycling becomes available, SuRe's cost could be significantly reduced. We added this discussion in Appendix E of the revised draft.

---

> ### Author Response · Authors · 2023-11-20
> **Response to Reviewer 9TBy (2/2)**
>
> **[W2 & Q1] The evaluation metrics are based on term overlaps and they might not capture all dimensions of model performance. Other factors like response coherence, relevance, and efficiency should also be considered … Why not consider LLMs approaches for automatic evaluation?**
>
> First, we would like to clarify that the considered EM/F1 scores are the common metrics for the considered ODQA datasets, even in the case of LLMs [5,6,7]. Therefore, we adopted those metrics for our experiments to make it easy to notice the significance of our results.
> Nevertheless, we agree that other factors are important metrics to be considered. Following your suggestion, we have conducted additional evaluations with LLMs approaches. Specifically, we measured two additional metrics: (1) MAUVE [8] and (2) LLM-acc [9].
>
> MAUVE is a recently proposed metric to compare the two distributions of the text generation model and human-written text using divergence frontiers. MAUVE (scale of 0 to 100, higher is better) is known for correlating highly with human judgments, and is frequently used to evaluate LMs’ responses [10,11]. LLM-acc assesses the accuracy (%) of LLMs’ responses to questions, using the prompting of LLMs instead of term overlap like EM/F1. We used the official code from the authors, only changing LLMs to ChatGPT. We measured this metric on NQ*, WebQ*, 2Wiki, and HotpotQA datasets, and the results are presented below.
>
> \begin{array}{c|cccc|c}
> \hline
> \text{MAUVE / LLM-acc} & \text{NQ*} & \text{WebQ*} & \text{2Wiki} & \text{HotpotQA} & \text{Average} \newline \hline
> \text{Base} & 81.3 / 53.2 & 61.3 / 48.8 & 35.1 / 36.2 & 62.4 / 51.6 & 60.0 / 47.5 \newline
> \text{SuRe (Ours)} & 95.9 / 56.2 & 75.7 / 51.4 & 52.2 / 48.2 & 89.6 / 52.4 & 78.3 / 52.1 \newline \hline
> \end{array}
>
> It's observed that our method significantly improves upon the baseline under these new evaluation metrics of LLMs approaches; this result further confirms that our framework enables LLMs to provide better answers to the given question. We have incorporated these results and the corresponding discussions in Appendix B.3 of the revised draft. We greatly appreciate your suggestion to strengthen our paper.
>
>
> **[Q2] Why only short-answer datasets? Have you considered long-answers? What would change in that case?**
>
> Thank you for the constructive suggestion. We agree that our approach could potentially be applied beyond short-answer datasets, and proving its effectiveness in broader applications would strengthen our paper. Accordingly, we have conducted further experiments on long-form question answering tasks to confirm the applicability of our approach.
>
> Specifically, we used the ASQA dataset [11,12], which comprises factoid questions and their corresponding long-form answers. For instance, the average length of the answers in the ASQA dataset is 71.8 words, compared to 2.6 words in the NQ dataset. For the experiments, we followed the setups in [11]; we compared the base prompting method with retrieval and SuRe on 948 test examples, using ChatGPT (GPT-3.5-turbo-0301) with 5 retrieved passages via GTR [13]. To evaluate, we measured ROUGE-L, String Exact Match (STR-EM) for correctness, and MAUVE [8] for fluency and coherence, in line with previous works [11,12]. The results are presented in the table below:
>
>
> \begin{array}{c|ccc}
> \hline
> \text{Methods / Metrics} & \text{ROUGE-L} & \text{STR-EM} & \text{MAUVE}  \newline \hline
> \text{Base} & 38.0 & 39.8 & 69.8 \newline
> \text{SuRe (Ours)} & 39.8 & 42.6 & 70.3 \newline \hline
> \end{array}
>
> These results show that our proposed framework consistently improves the performance of retrieval-augmented LLMs for long-form QA tasks. However, we acknowledge that there is still room for improvement, particularly in finding better prompt designs, given that our current designs are based on performance on short-answer datasets. We hope future research will explore this direction, extending the benefits of our framework to broader QA scenarios with LLMs. We have included these results and the corresponding discussions in Appendix B.4 of the revised draft.
>
> ---
>
>
> If you have any further questions/concerns, please do not hesitate to let us know.
>
> Thank you very much,
> Authors

---

> ### Author Response · Authors · 2023-11-20
> **References**
>
> [1] Nogueira and Cho, Passage Re-ranking with BERT., arXiv:1901
> [2] Ren et al., RocketQAv2: A Joint Training Method for Dense Passage Retrieval and Passage Re-ranking., EMNLP 2021
> [3] Das et al., Multi-step Retriever-Reader Interaction for Scalable Open-domain Question Answering., ICLR 2019
> [4] Trivedi et al., Interleaving Retrieval with Chain-of-thought Reasoning for Knowledge-intensive Multi-step Question., ACL 2023
> [5] Chowdhery et al., PaLM: Scaling Language Modeling with Pathways., arXiv:2204
> [6] Izacard et al., Atlas: Few-shot Learning with Retrieval Augmented Language Models., JMLR 2023
> [7] Shi et al., REPLUG: Retrieval-Augmented Black-Box Language Models., EMNLP 2023 Findings
> [8] Pillutla et al., MAUVE: Measuring the Gap Between Neural Text and Human Text using Divergence Frontiers., NeurIPS 2021
> [9] Kamalloo et all., Evaluating Open-Domain Question Answering in the Era of Large Language Models., ACL 2023
> [10] Su et al., A Contrastive Framework for Neural Text Generation., NeurIPS 2022
> [11] Gao et al., Enabling Large Language Models to Generate Text with Citations., EMNLP 2023
> [12] Stelmakh et al., ASQA: Factoid Questions Meet Long-Form Answers., EMNLP 2022
> [13] Ni et al., Large Dual Encoders Are Generalizable Retrievers., EMNLP 2022

---

### Official Review · Reviewer_Qrw6 · 2023-11-07

**Soundness:** 3 good
**Presentation:** 3 good
**Contribution:** 3 good
**Rating:** 6
**Confidence:** 4

**Summary:**

This paper proposes a method to improve open-domain QA by making a summary of the retrieved passages. To create good summaries, candidates are first generated by an LLM, which are then used to condition the generation of summarties. The method is compared with a naive augmentation with all the retrieved passages and a generic summarization, as well as several existing approaches such as reranking, CoT, etc. The proposed method is shown to perform better on several datasets.

**Strengths:**

The idea of creating a summary of the retrieved passages centered around the possible answers is very interesting. This may solve the problem of noise information contained in the passages and help dealing with long passages. This idea has not been explored previously.
The approach relies on prompts to LLM, so it can be used with any LLM without fine-tuning it. This may be a generally feasible approach in many application contexts.
The experimental results are convincing. It demonstrates the a answer-oriented summary is better than a generic summary, and better than no summarization. The advantage of the approach is properly shown. In addition, the method is also shown to outperform the existing methods.

**Weaknesses:**

The performance of the method may strongly depend on the prompts used. While the paper demonstrates that appropriate prompts can help create a good summary for improving QA, there are still questions about what prompts should be used. I wonder if the authors have tested several alternative prompts before choosing the ones used.
Fig 3 is unclear. What are "the corresponding two conditional summarizations"?

**Questions:**

See comments in Weakness.

---

> ### Author Response · Authors · 2023-11-20
> **Response to Reviewer Qrw6**
>
> Dear Reviewer Qrw6,
>
> We sincerely appreciate your review with thoughtful comments. We have carefully considered each of your questions and provide detailed responses below. Please let us know if you have any further questions or concerns.
>
> ---
>
> **[W1] The performance of the method may strongly depend on the prompts used … I wonder if the authors have tested several alternative prompts before choosing the ones used.**
>
> Yes, we examined several prompt designs before finalizing those used in the experiments. We'd like to share two key observations from this procedure for the reviewer's interest.
> First, precise and detailed instructions are crucial. As each component of the proposed framework operates in a zero-shot manner, its output greatly relies on the provided instruction. For example, in answer candidate generation (Eq.2), the current prompt (from Appendix A.2 and presented below) consistently outperforms the initially considered "previous" prompt.
> - (**Current**) *Below are N passages related to the question at the end. After reading the passages, provide two correct candidates for the answer to the question at the end. Each answer should be in the form: (a) xx, (b) yy.)*
> - (**Previous**) *Task description: give two candidates for the answer to the following question (e.g., (a) xx, (b) yy).*
>
> Second, proper input arguments are essential. For instance, along with the target candidate, providing all candidates as additional input enhanced the quality of conditional summarization. This is because it helps to further specify which contexts of retrieval should be the focus. However, including this information, or even the retrieval passages, disrupted the verification step by interrupting the focus on the summarizations.
>
> Despite these efforts, we believe there is still potential for further improvement through better prompt designs. We remain this as a future work and hope our observations can provide insights to other researchers. We have added this discussion to Appendix A.8 of the revised draft. Thank you for your insightful comments.
>
> ---
>
> **[W2] Fig 3 is unclear. What are "the corresponding two conditional summarizations"?**
>
> “The corresponding two conditional summarizations” indicate that two generated summarizations to support two different answer candidates accordingly; for example, summarization #1 is generated to support candidate #1. Therefore, Fig. 3 implies that the context of each generated summarization is specialized to the given answer candidate. Consequently, the generated summarization exhibits a higher similarity with the corresponding answer candidate than the other answer candidate. To clarify this, we revised the relevant paragraph for Fig. 3 as follows:
> - (**Original**) Also, we verify that the generated contexts significantly vary between different candidates, when we measure the text similarity based on TF-IDF between two candidates and corresponding two conditional summarizations on the Natural Question dataset, as shown in Figure 3.
> - (**Revised**) Also, we verify that the contexts of the generated summarization are specialized on a given answer candidate; when we measure TF-IDF based text similarity between two candidates and two conditional summarizations from each candidate (e.g., summarization \#1 is generated to support answer candidate \#1) on Natural Question dataset in Figure 3, the summarization exhibits a higher similarity with the corresponding candidate than the other candidate.
>
> ---
>
> If you have any further questions/concerns, please do not hesitate to let us know.
>
> Thank you very much,
> Authors

---

> > ### Comment · Reviewer_Qrw6 · 2023-11-21
> >
> > Thanks for the answers in the rebuttal. This clarified some questions.

---

> > > ### Author Response · Authors · 2023-11-23
> > > **Response to Reviewer Qrw6**
> > >
> > > Dear Reviewer Qrw6,
> > >
> > > We are glad to see that our rebuttal successfully clarified your question! Please let us know if you have any further questions.

---

### Official Review · Reviewer_mMA8 · 2023-11-08

**Soundness:** 3 good
**Presentation:** 3 good
**Contribution:** 3 good
**Rating:** 6
**Confidence:** 4

**Summary:**

This work studies retrieval augmentation via prompting, for the task of open-domain question answering. They propose a method based on "summarized retrieval" (SuRe). SuRe proceeds in a few steps. First, it retrieves top-k results with an off-the-shelf retriever. Second, it generates multiple candidate answers directly (not via decoding multiple completions). Third, it generates a conditional summary (of the retrievals) for each candidate answer. Lastly, it validates each summary (as faithful or not) and uses a pairwise comparison approach to select the most informative answer. The pairwise scoring approach is applied across all pairs and averaged.

On the Open-Domain QA tasks NQ, WebQ, 2Wiki, and HotpotQA, the authors report improvements of up to 4.4% in exact match over baselines. The authors conduct a human evaluation of the SuRe summaries (whose defining characteristic is being centered around a candidate answer, derived from GPT-4) against general-purpose summaries (derived from GPT-4 without answer candidates). Generic summarization wins 30.3% while SURE wins 37.4%. They ask human evaluators which summaries are more informative and better support the question-answer pairs, and observe higher preference for SuRe (Generic: 26.9% vs SuRe: 43.4%).

**Strengths:**

1. The proposed pipeline is relatively rich and well-executed.

2. The authors develop a number of thoughtful baselines and report extensive comparisons. While there's very limited comparisons to prior work directly, I do find that there's a lot of value in the set of curated baselines they develop, which can be compared apples to apples to the proposed method.

3. The results are consistently solid across several tasks, LMs, and retrievers. This is the hallmark of a solid idea. The results are never that strong overall (in isolation), but perhaps SuRe can probably be combined into a really strong 'sota' system in principle.

**Weaknesses:**

1. The authors assert in the abstract that retrieval augmentation via prompting "has been limitedly explored". While much more work is required to improve RAG methods that use prompting (or otherwise), few areas of modern NLP that have received more attention than RAG prompting. As a case in point, the authors build a method for "summarized retrieval", but I don't see citations to much prior work on considering summarization in the context of open-domain QA and prompting. For example, "Baleen: Robust Multi-Hop Reasoning at Scale via Condensed Retrieval" is the title of a paper at NeurIPS 2021, where the notion of _condensed retrieval_ seems fundamentally connected to _summarized retrieval_. For another example, "Open Domain Multi-document Summarization: A Comprehensive Study of Model Brittleness under Retrieval" is a recent task proposal. These are certainly different formulations of summarization at scale, but they are just two examples of a rich space considering summarization in open-domain contexts.


2. The authors focus on 'zero-shot prompting', but I do not find a convincing justification for presenting this limitation as a 'remarkable' feature. Zero-shot prompts are not necessarily indicative of generality (if anything, a decent few-shot prompt specifies the task more precisely and is empirically not unlikely to be more robust across LMs, counter to the assertion by the authors). While I'm not opposed to the need to eliminate some angles from a large experimental endeavor, I do wonder how useful SuRe is if the QA component had access to a few examples of the task. (This overall may explain, for instance, why chain of thought performs so poorly in the evaluations.)

**Questions:**

See weaknesses.

---

> ### Author Response · Authors · 2023-11-20
> **Response to Reviewer mMA8**
>
> Dear Reviewer mMA8,
>
> We sincerely appreciate your review with thoughtful comments. We have carefully considered each of your questions and provide detailed responses below. Please let us know if you have any further questions or concerns.
>
> ---
>
> **[W1] The authors assert in the abstract that retrieval augmentation via prompting "has been limitedly explored" ... but I don't see citations to much prior work on considering summarization in the context of open-domain QA and prompting ... These are certainly different formulations of summarization at scale, but they are just two examples of a rich space considering summarization in open-domain contexts.**
>
> We appreciate your comment about the additional related works and their relevance to the proposed framework. As you pointed out, there are some prior works that consider summarizations of retrieved documents in open-domain contexts, including the referred works. However, we first clarify that our work is certainly distinct from those works.
>
> - (**Original**) First, unlike the generic summarization used in the existing works, our framework newly introduces conditional summarization, which is specified to support the given answer candidate. In addition, we further incorporate the verification step to determine which answer the candidate should be chosen, by comparing and evaluating these conditional summarizations. With these components, our framework significantly outperforms LLMs using the generic summarization (see Table 3). In addition, it is noteworthy that we demonstrated the advantages of our conditional summarizations over the generic ones through extensive experiments (see Table 4 and Figure 4). These results indicate that our core contribution is not just introducing the summarization, but proposing a new framework with the proposed components.
>
> Nevertheless, we agree that our draft could be more solid by adding more related works including the suggested ones, and specifying their limitations. To this end, we added the following paragraph to the related work section, in our revised draft:
>
> - (**Revised**) … On the other hand, the summarization of retrieved passages has been considered in open-domain context; for example, there are recent works that propose to learn a module to selectively use the retrieved information in sentence- (Khattab et al., 2021; Su et al., 2022) or passage-level (Mao et al., 2021; Chuang et al., 2023). In addition, Su et al. (2022); Giorgi et al. (2023) formed a new task that combines both passage retrieval and summarization for a given query, and Gao et al. (2023) consider summarization of information for prompting. However, these works require a large annotated dataset to extract the information specified to answer the question or construct the generic summarization which focuses on preserving the retrieved information within reduced texts.
>
> Thank you very much for the suggestion to strengthen our paper.
>
> ---
>
> **[W2] I do wonder how useful SuRe is if the QA component had access to a few examples of the task.**
>
> Thank you for the constructive suggestion. We agree that demonstrating our framework with few-shot examples would further strengthen our paper. Following your suggestion, we have conducted additional experiments on NQ* and WebQ* datastets, using 1-shot and 5-shot examples from training datasets. We measured the average EM/F1 of base prompting with 10 retrieved passages and SuRe, across four different random seeds used for sample selection.
>
> The tables below show that adding few-shot examples improves QA accuracy for both the baseline and SuRe. Specifically, we observed that SuRe's gain primarily results from generating more accurate answer candidates. These findings suggest that our proposed method could be effective in scenarios beyond the zero-shot setup considered. Therefore, we believe that our work could contribute to broader ODQA scenarios in the future. We have included these results and related discussions in Appendix B.5 of the revised draft.
>
>
> \begin{array}{c|ccc}
> \hline
> \text{Dataset: NQ*} & \text{0-shot (EM/F1)} &\text{1-shot (EM/F1)} & \text{5-shot (EM/F1)} \newline \hline
> \text{Base} & 29.4 / 41.7 & 30.1 / 39.3 & 31.9 / 42.0 \newline
> \text{SuRe (Ours)} & 35.6 / 44.9  & 36.3 / 46.8 & 37.2 / 47.7   \newline \hline
> \end{array}
> \begin{array}{c|ccc}
> \hline
> \text{Dataset: WebQ*} & \text{0-shot (EM/F1)} &\text{1-shot (EM/F1)} & \text{5-shot (EM/F1)} \newline \hline
> \text{Base} & 19.4 / 32.2 & 19.6 / 32.9 & 19.9 / 33.5  \newline
> \text{SuRe (Ours)} &23.2 / 36.5  & 24.2 / 39.4 & 24.3 / 38.5  \newline \hline
> \end{array}
>
> ---
>
> If you have any further questions/concerns, please do not hesitate to let us know.
>
> Thank you very much,
> Authors

---

> > ### Comment · Reviewer_mMA8 · 2023-11-23
> >
> > Thank you for the detailed response. I believe you have addressed my feedback, and had there been a score 7/10 I'd be willing to raise my score to that. (Unfortunately, the system only supports incrementing from 6/10 to 8/10. Nonetheless, I will champion this work at the level of 7/10 in reviewer discussions.)

---

> > > ### Author Response · Authors · 2023-11-23
> > > **Response to Reviewer mMA8**
> > >
> > > Dear Reviewer mMA8,
> > >
> > > We appreciate that our rebuttal addressed your concerns. Also, thank you for the support for our work! Please let us know if you have any further questions.

---

### Author Response · Authors · 2023-11-20
**General Response**

Dear reviewers and AC,

We sincerely appreciate your valuable time and effort spent reviewing our manuscript.

As reviewers highlighted, we propose a simple (9TBy), yet novel (AgBt, Qrw6) method for open-domain QA with LLMs, that shows strong empirical results (AgBt, 9TBy, Qrw6, mMA8) on the comprehensive experiments (MmoS, AgBt) with clear writing (MmoS, AgBt).

We appreciate your constructive feedback on our manuscript. In response to the comments, we have carefully revised and enhanced the manuscript, including the following additional discussions and experiments:
- Further clarify our contribution (Title and abstract)
- Detailed explanation about the limitations of previous works (Sections 1 and 2)
- More detailed descriptions for Figure 3 (Section 3.2)
- Clarify the mapping between Table 3 and each component of our proposed method (Table 3 and Section 4.3)
- Better visualization and caption for figures (Figure 4 and Section 4.3)
- Additional discussion and demonstration of the used prompts (Appendix A.7 and A.8)
- More experimental results with SuRe (Appendix B.3, B.4, and B.5)
- Discussion on cost and quality gain (Appendix E)

In the revised manuscript, these updates are temporarily highlighted in "green” for your convenience to check. We sincerely believe that these updates may help us better deliver the benefits of the proposed SuRe to the ICLR community.

Thank you very much,
Authors.

---

### Meta-Review · Area_Chair_r2BX · 2023-12-06

**Metareview:**

This paper presents a method for summarized retrieval: retrieved passages for multiple answer candidates are summarized, then evaluated for validity and ranked. Specifically, the pipeline involves (1) retrieval; (2) answer candidate generation from the retrieved passages; (3) conditional summarization (summarizing the passages for a specific answer); (4) verification.  The experiments explore the method's effectiveness in a variety of settings.

The reviewers generally liked the idea of this pipeline. It is flexible and generalizes to different LLMs.  The reviewers found the results to be "solid", a characterization I agree with.

The relation to prior work was brought up by mMA8 and addressed in revision; I think that while similar pipelines have been explored, the interest in this task is high, so there is room in the literature to flesh out points in the design space here.

Scalability was raised as well, though I think the authors effectively address this by arguing about the modularity and increased effectiveness of the pipeline. They also discuss cost.

Finally, framing in the title/abstract was changed, and I think these improve the paper from the submitted version.

**Justification For Why Not Higher Score:**

This idea is nice, but similar ideas are floating around in other papers; I think this paper is well-executed but not exceptionally groundbreaking. The reviews reflect that it is solid but few reviewers are *really* excited about it.

**Justification For Why Not Lower Score:**

The authors addressed all the weaknesses and the reviewers are all positive.

---

### Decision · Program_Chairs · 2024-01-16

Accept (poster)